# Probability Signature: Bridging Data Semantics and Embedding Structure in Language Models

## Abstract

The embedding space of language models is widely believed to capture the semantic relationships; for instance, embeddings of digits often exhibit an ordered structure that corresponds to their natural sequence. However, the mechanisms driving the formation of such structures remain poorly understood. In this work, we interpret the embedding structures via the token relationships. We propose a set of probability signatures that reflect the semantic relationships among tokens. Through experiments on the composite addition tasks using the linear model and feedforward network, combined with theoretical analysis of gradient flow dynamics, we reveal that these probability signatures significantly influence the embedding structures. We further generalize our analysis to large language models (LLMs). Our results show that the probability signatures are faithfully aligned with the embedding structures, particularly in capturing strong pairwise similarities among embeddings. Our work offers a universal analytical framework that investigates how token relationships direct embedding geometries, empowering researchers to trace how gradient flow propagates token relationships onto embedding structures of their models.

## 1 Introduction

In recent years, deep neural network-based large language models (LLMs) have demonstrated remarkable performance (Comanici et al., 2025; OpenAI et al., 2024; DeepSeek-AI et al., 2025). The development of these models has largely followed what Richard Sutton termed "the bitter lesson"–that the most effective approach to improving AI performance has historically been to leverage greater computational resources, larger models, and more data, rather than incorporating human knowledge or specialized architectures (Sutton, 2019). This trend has been formalized through scaling laws (Kaplan et al., 2020). While these scaling laws provide valuable quantitative predictions for model performance, they also reveal a concerning limitation: achieving further significant improvements may require prohibitively large increases in model and data size, making continued scaling increasingly impractical and resource-intensive.

A more sustainable path forward lies in developing a mechanistic understanding of deep learning's success. Recent research has uncovered key properties such as the edge-of-stability phenomenon (Wu et al., 2018; Cohen et al., 2021), frequency principle (Xu et al., 2020; 2025a), attention patterns (Elhage et al., 2021; Olsson et al., 2022; Bhojanapalli et al., 2020), and parameter distribution characteristics (Kovaleva et al., 2021; Dar et al., 2023). Among these, the structure of the embedding space is fundamental: it serves as the gateway through which tokens are encoded, forming the basis of all subsequent learning. Indeed, embeddings often capture intuitive semantics—for instance, embeddings of digits 1,2,...,9 form an ordered structure reflecting their numerical sequence (Mikolov et al., 2013b; Ethayarajh et al., 2019; Zhang et al., 2024; Yao et al., 2025). Yet, what drives this alignment between embedding geometry and semantic structure remains an open question: the precise mechanisms linking data distribution to embedding organization are still poorly characterized.

In this work, we establish a mechanistic link between embedding geometry and token relationship through the lens of gradient flow dynamics. For each token, we propose a set of probability signa-

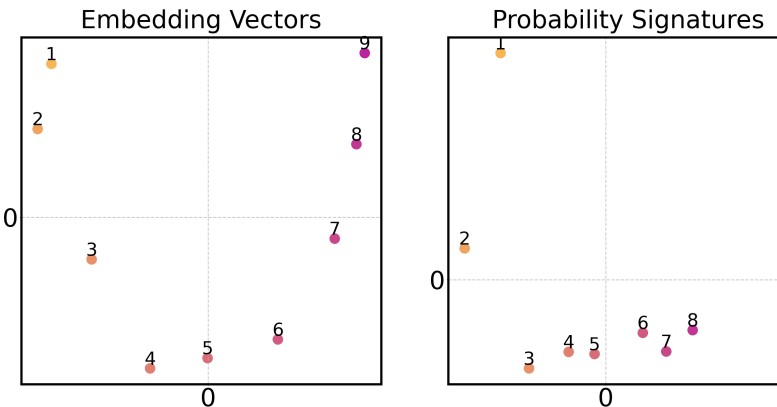

Figure 1: Left: The PCA projection of embedding vectors of the digits $1, 2, 3, \cdots, 9$ in Qwen2.5 3B-base. Right: The PCA projection of the probability signatures of the digits $1, 2, 3, \cdots, 9$ estimated by subsets of Pile(detailed formulation see (4)).

tures based on its statistical relationships with the other tokens (e.g., label distribution, co-occurrence patterns). Such probability signatures systematically capture inherent token-level relationships and reflect semantic structures. Our gradient flow analysis reveals that these signatures actively govern the evolution of embedding vectors, forging a deterministic connection between probability signature and embedding structure. This is illustrated in Figure 1: both the embeddings of digits 1,2,...,9 in Qwen2.5 3B-base (Team, 2024) and their probability signatures estimated from the Pile corpus (Gao et al., 2020; Biderman et al., 2022) exhibit an ordered arrangement aligned with their natural sequence, suggesting that probability signatures are the prime driver of embedding organization. We instantiate this framework by deriving the exact signature sets for linear models and feedforward networks, showing how architecture determines which token relationships are encoded. Through carefully controlled synthetic tasks, we verify that manipulating probability signatures predictably reshapes the embedding space. Finally, we extrapolate our framework to LLMs, demonstrating that even in realistic training regimes, next-token and previous-token distributions dominate the dynamics of embedding and unembedding vectors in Qwen2.5 and Llama-2 architectures.

The primary contribution of this work is a universal analytical framework that investigates how token relationships direct embedding geometries. Through exact gradient flow analysis, we demonstrate that any embedding-based architecture encodes a specific, predictable subset of data distribution statistics into its token representations. This framework not only explains observed embedding structures as a deterministic consequence of probability signatures, but also predicts which probability signatures dominate in a given model, transforming representation learning from a black-box phenomenon into a transparent, distribution-driven process.

## 2 RELATED WORK

**Parameter analysis in LLMs** Investigating the underlying parameter properties in LLMs is crucial for understanding the foundation of models. Some works focus on the specific modules in models. Elhage et al. (2021); Olsson et al. (2022) uncover mechanisms such as induction heads from the attention module. Bhojanapalli et al. (2020) reveals the rank-collapse phenomenon of the attention matrix. Geva et al. (2021; 2022); Dai et al. (2022) investigates the characteristics and functions of the FFN in LLMs. Additionally, analysis of a single neuron has been widely employed in mechanism interpretation, particularly in circuits analysis Hanna et al. (2023); Wang et al. (2023); Hanna et al. (2024); Wang et al. (2025), sparse autoencoders (SAE) Huben et al. (2024); Bricken et al. (2023), transcoders Dunefsky et al. (2024), and cross-layer transcoders (CLT) Ameisen et al. (2025). There are also some studies investigating the global properties of all parameters. Dar et al. (2023); Katz et al. (2024) introduce a framework for interpreting all parameters of Transformer models by projecting them into the embedding space. Kovaleva et al. (2021); Yu et al. (2025) provide an analysis of the parameter distribution, demonstrating the significance of these outliers. In this work, we will focus on the embedding space, explaining the formation of its structure from both experimental and theoretical perspectives.

**Embedding structure and representation learning** Since the introduction of static word embeddings by Mikolov et al. (2013a); Pennington et al. (2014) and the adoption of contextualized embeddings (Devlin et al., 2019; Peters et al., 2018), significant attention has been devoted to analyzing embedding properties. Gao et al. (2019); Ethayarajh (2019); Timkey & van Schijndel (2021) explore the anisotropy of embedding space, while Cai et al. (2021) show that embeddings exhibit isotropy within clusters. Liu et al. (2022) offers insights into grokking by emphasizing the role of well-organized embedding structures. Zhang et al. (2024) establishes a connection between embedding structure and model generalization, and Yao et al. (2025) provides an analysis of this relationship. Crucially, these studies characterize embedding geometry post hoc, treating it as an empirical phenomenon to be observed rather than a deterministic outcome to be explained. In contrast, we mechanistically interpret how embedding structures arise from token relationships. Our gradient-flow-driven framework reveals that token-wise probability signatures dictate the evolution of embedding vectors, offering not merely a new perspective, but a predictive, architecture-agnostic protocol for understanding representation formation.

## 3 PRELIMINARY

### 3.1 EMBEDDING-BASED MODEL

We denote the models functioning on the trainable embedding of the input sequence as embedding-based models. We provide the following formulation:

**Definition 1.** *Given a vocabulary $\mathcal{V} \subset \mathbb{N}^+$ with size $d_{\mathrm{vob}}$, we denote $e_x \in \mathbb{R}^{d_{\mathrm{vob}}}$ as the one-hot vector of $x$ for any $x \in \mathcal{V}$. The trainable embedding matrix and unembedding matrix are $W^E \in \mathbb{R}^{d \times d_{\mathrm{vob}}}$ and $W^U \in \mathbb{R}^{d_{\mathrm{vob}} \times d}$, respectively. For a sequence $X := [x_1, x_2, \cdots, x_L] \in \mathcal{V}^L$ with length $L$. The trainable embedding of $X$ and an embedding-based model $F$ taking $X$ as input could be formulated as*

$$W_X^E = W^E e_X := \left[ W_{x_1}^E, W_{x_2}^E, \cdots, W_{x_L}^E \right],$$
$$F(X) = W^U G\left( W_X^E \right),$$

*where $G$ means the mapping in the hidden space, $W_{x_i}^E = W^E e_{x_i}$ represents the embedding vector of elements $x_i \in X$.*

Embedding-based models have been widely applied in various domains, particularly in NLP. In this work, our objective is to investigate how the token relationships impact the characteristics of the embedding space. We will begin with the following simplified models, facilitating our analysis.

- Linear model. $F_{\mathrm{lin}}(X) = W^U \sum_{x \in X} W_x^E$.
- Feedforward network. $F_{\mathrm{ffn}}(X) = W^U \sigma\left( \sum_{x \in X} W_x^E \right)$, where $\sigma$ denotes the element-wise nonlinear activation.

Furthermore, we will provide an elementary analysis of the Transformer architecture in language tasks and verify our results by the Qwen2.5 architecture and the Llama 2 architecture (Touvron et al., 2023).

### 3.2 TOKEN RELATIONSHIPS & PROBABILITY SIGNATURES

In natural language, a token's meaning is fully constituted by its statistical context: how it predicts downstream labels, what tokens it co-occurs with, and how these relationships jointly evolve. Formally, these semantic regularities manifest as conditional probability distributions over the vocabulary. Denote the label of a sequence $X$ by $y$ and assume $(X, y) \sim \pi$. For a token $x$ in input $X$, we consider four representative families of such distributions:

- **Label relationship**: $\mathbb{P}_\pi(y = \nu \mid x \in X)$ encodes what $x$ signals about the output—e.g., "excellent" in a review robustly predicts a positive label $\nu$, while "frustrated" skews toward negative.
- **Co-occurrence relationship**: $\mathbb{P}_\pi(x' \in X \mid x \in X)$ captures syntactic-semantic neighborhoods—"stock" frequently co-occurs with "market" but rarely with "apple" (in the financial sense). Higher-order terms like $\mathbb{P}_\pi(x', x'' \in X \mid x \in X)$ encode compositional contexts.

- **Joint relationship**: The joint $\mathbb{P}_\pi(x' \in \boldsymbol{X}, y = \nu \mid x \in \boldsymbol{X})$ reveals context-dependent labeling—"apple" co-occurring with "pie" predicts a food label, while with "store" predicts a tech label.

- **Inverse relationship**: $\mathbb{P}_\pi(x_i \in \boldsymbol{X} \mid y = x)$ describes what precedes a token as its cause—the tokens that predict $x$ itself (e.g., what contexts make "surprised" likely to appear).

These token-wise relationships are semantic primitives: they are computable from data, independent of any model, but depend on the contexts and tokenizers, yet fully determine the token's functional role in the corpus. Critically, a sequence of length $L$ yields exponentially many such relationships—our four families merely scratch the surface. **Rather than exhaustively enumerating them, we propose a systematic principle: the gradient flow dynamics of any embedding-based model will automatically select a specific subset of these relationships to encode.** To showcase this principle, we distill each family into a compact **probability signature**—a vector/matrix that aggregates the relevant conditional probabilities (Definition 2). This choice is deliberate: we aim not to prescribe a fixed signature set, but to demonstrate that any such set derived from gradient flow analysis will faithfully sculpt the embedding space.

**Definition 2** (Probability Signatures). *For token $x \in \mathcal{V}$, we define four probability signatures that capture distinct token relationships:*

$$\boldsymbol{\phi}_x^y = \sum_{\nu \in \mathcal{V}} \mathbb{P}_\pi(y = \nu \mid x \in \boldsymbol{X}) \boldsymbol{e}_\nu, \qquad \boldsymbol{\phi}_x^{\boldsymbol{X}} = \sum_{x' \in \mathcal{V}} \mathbb{P}_\pi(x' \in \boldsymbol{X} \mid x \in \boldsymbol{X}) \boldsymbol{e}_{x'},$$

$$\boldsymbol{\phi}_x^{\boldsymbol{X}|y} = \sum_{\nu,x'} \mathbb{P}_\pi(x' \in \boldsymbol{X}, y = \nu \mid x \in \boldsymbol{X}) \boldsymbol{e}_\nu \times \boldsymbol{e}_{x'}^\top, \qquad \boldsymbol{\varphi}_x^{\boldsymbol{X}} = \sum_{x' \in \mathcal{V}} \mathbb{P}_\pi(x' \in \boldsymbol{X} \mid y = x) \boldsymbol{e}_{x'}.$$

*We have $\boldsymbol{\phi}_x^y, \boldsymbol{\phi}_x^{\boldsymbol{X}}, \boldsymbol{\varphi}_x^{\boldsymbol{X}} \in \mathbb{R}^{d_{\mathrm{vob}}}, \boldsymbol{\phi}_x^{\boldsymbol{X}|y} \in \mathbb{R}^{d_{\mathrm{vob}} \times d_{\mathrm{vob}}}$.*

Each probability signature is a data-derived feature vector/matrix for $x$. For example, the $\nu$-th element of $\boldsymbol{\phi}_x^y$ is $\mathbb{P}_\pi(y = \nu \mid x \in \boldsymbol{X})$. The signatures above are exemplars; our framework empowers researchers to derive more probability signatures for their models of interest by tracing how gradient flow propagates token relationships onto embedding structures.

# 4 GRADIENT FLOW OF EMBEDDING VECTOR

To understand why embeddings organize as they do, we examine the continuous dynamics of training via gradient flow, the limit of gradient descent as the learning rate vanishes. This tool acts as a microscope, revealing the "force field" that sculpts each embedding vector. Formally, Given a dataset $\{(\boldsymbol{X}^i, y^i)\}_{i=1}^N$ with loss function $\ell^i = \ell(F(\boldsymbol{X}^i; \theta), y^i)$, the gradient descent implies that $\theta^{k+1} - \theta^k = -\eta \frac{1}{N} \sum_{i=1}^N \frac{\partial \ell^i}{\partial \theta} \mid_{\theta = \theta^k}$. Then the gradient flow of $\theta$ is defined as:

$$\frac{d\theta}{dt} := \lim_{\eta \to 0} \frac{\theta^{k+1} - \theta^k}{\eta} = -\frac{1}{N} \sum_{i=1}^N \frac{\partial \ell^i}{\partial \theta}.$$

Our goal is to trace how this dynamics acts on the embedding vector $\boldsymbol{W}_x^E$ for any token $x \in \mathcal{V}$. Using the standard cross-entropy loss:

$$\ell^i = -\log \mathrm{Softmax}(F(\boldsymbol{X}^i))_{y^i} = -\log \frac{\exp F(\boldsymbol{X}^i)_{y^i}}{\sum_{j=1}^{d_{\mathrm{vob}}} \exp F(\boldsymbol{X}^i)_j},$$

we derive the exact evolution equation:

**Proposition 1.** *Let $\odot$ represent the Hadamard product and $T$ mean the matrix transpose. Given an embedding-based model $F$ with an embedding matrix $\boldsymbol{W}^E$. For any token $x \in \mathcal{V}$, the gradient flow of $\boldsymbol{W}_x^E$ (the embedding vector of $x$) can be formulated as follow when $N \to \infty$:*

$$\frac{d\boldsymbol{W}_x^E}{dt} = r_x^{\mathrm{in}} \left( \sum_{\nu \in \mathcal{V}} \mathbb{P}_\pi(y = \nu \mid x \in \boldsymbol{X})(\boldsymbol{W}^{U,T} \boldsymbol{e}_\nu) \odot \mathbb{E}_\pi\left[ G^{(1)}(\boldsymbol{W}_{\boldsymbol{X}}^E) \mid x \in \boldsymbol{X}, y = \nu \right] \right.$$

$$\left. -\mathbb{E}_\pi\left[ (\boldsymbol{W}^{U,T} \boldsymbol{p}) \odot G^{(1)}(\boldsymbol{W}_{\boldsymbol{X}}^E) \mid x \in \boldsymbol{X} \right] \right)$$

$$:= r_x^{\mathrm{in}} \left( \boldsymbol{U} \boldsymbol{\phi}_x^y - \mathbb{E}_\pi\left[ (\boldsymbol{W}^{U,T} \boldsymbol{p}) \odot G^{(1)}(\boldsymbol{W}_{\boldsymbol{X}}^E) \mid x \in \boldsymbol{X} \right] \right),$$

where $U \in \mathbb{R}^{d \times d_{\text{vob}}}$ and the $\nu$-th column of $U$ equals $\left(W^{U,T}e_{\nu}\right) \odot \mathbb{E}_{\pi}\left[G^{(1)}\left(W_{\boldsymbol{X}}^{E}\right) \mid x \in \boldsymbol{X}, y = \nu\right]$. $r_x^{\text{in}}$ denotes the ratio of input sequences containing $x$ in the training set, $G^{(1)}$ represents the derivative of $G$ with respect to $\boldsymbol{W}_x^E$ and $\boldsymbol{p} = \text{softmax}\left(F\left(\boldsymbol{X}\right)\right)$.

This equation reveals that $\phi_x^y$ drives $\boldsymbol{W}_x^E$ toward a direction determined by the token-label semantics. This means: if two tokens share similar label distributions, their embeddings will be forced to evolve in similar directions from the very start of training. The emergence of other probability signatures ($\phi_x^{\boldsymbol{X}}$, $\phi_x^{\boldsymbol{X}|y}$) is dependent on the formulation of $G$, as we will show next.

To make this analysis concrete, we dissect linear model and feedforward networks, deriving their exact probability signature sets from Proposition 1. This demonstrates how our framework systematically extracts the relevant probability signatures for any given $G$.

### 4.1 Linear Model

For linear models $F_{\text{lin}}$, the hidden mapping $G$ is simply the sum of embeddings. Substituting this into Proposition 1 yields a simplified dynamics where the gradient flow depends on only two probability signatures:

**Corollary 1** (Embedding of Linear Model). *Let $N \to \infty$, $\pi$ denotes the data distribution over the training set. The gradient flow of $\boldsymbol{W}_x^E$ in $F_{\text{lin}}$ can be approximated by*

$$\frac{d\boldsymbol{W}_x^E}{dt} = r_x^{\text{in}}\boldsymbol{W}^{U,T}\left(\phi_x^y - \frac{1}{d_{\text{vob}}}\boldsymbol{W}^U\boldsymbol{W}^E\phi_x^{\boldsymbol{X}} + \boldsymbol{\eta}\right), \tag{1}$$

*where $\boldsymbol{\eta}$ denotes the data-independent and higher-order terms.*

The Corollary 1 indicates that the term $\phi_x^y$ acts as the primary steering force. Early in training, when $\|\boldsymbol{W}^U\boldsymbol{W}^E\|$ is small, $\phi_x^y$ alone dictates the update direction. The term $\phi_x^{\boldsymbol{X}}$ modulates the embedding update based on contextual co-occurrence statistics, but its influence is scaled by $\frac{1}{d_{\text{vob}}}\boldsymbol{W}^U\boldsymbol{W}^E$ and thus emerges later in training.

**Experimental Validation: Controllable Addition Tasks** If two tokens $\alpha, \alpha'$ satisfy $\phi_\alpha^y \approx \phi_{\alpha'}^y$ and $\phi_\alpha^{\boldsymbol{X}} \approx \phi_{\alpha'}^{\boldsymbol{X}}$, Corollary 1 forces their embeddings to align: $\cos(\boldsymbol{W}_\alpha^E, \boldsymbol{W}_{\alpha'}^E) = \frac{\boldsymbol{W}_\alpha^{E,T}\boldsymbol{W}_{\alpha'}^E}{||\boldsymbol{W}_\alpha^E||_2||\boldsymbol{W}_{\alpha'}^E||_2} \to 1$. We design three **variable-controlled addition tasks** to isolate and verify each probability signature's influence. In each task, $\phi_\alpha^y$ or $\phi_\alpha^{\boldsymbol{X}}$ or both of them will be identical across $\alpha$. Assuming all tokens belong to positive integers, and we denote an anchor set by $\mathcal{A}$, whose elements represent different addition operations, i.e., anchor $\alpha_1$ means addition with $\alpha_1$. Given a input sequence $\boldsymbol{X} = [z, \alpha_1, \alpha_2]$, we define the following tasks:

- **Addition task** (Varying $\phi_\alpha^y$). $y = f_{\text{add}}\left(\boldsymbol{X}\right) = z + \alpha_1 + \alpha_2, \quad \alpha_1, \alpha_2 \in \mathcal{A}$. For each anchor pair $(\alpha_1, \alpha_2)$, $z$ is sampled from the same set $\mathcal{Z}$ with $\mathcal{Z} \cap \mathcal{A} = \emptyset$. In this task, $\phi_\alpha^{\boldsymbol{X}}$ are identical across anchors while $\phi_\alpha^y$ are distinct with varying anchors $\alpha$.

- **Addition task with the same value domain** (Varying $\phi_\alpha^{\boldsymbol{X}}$). $y = \tilde{f}_{\text{add}}\left(\boldsymbol{X}\right) = z + \alpha_1 + \alpha_2, \quad \alpha_1, \alpha_2 \in \mathcal{A}$. For anchor pair $(\alpha_1, \alpha_2)$, $z \in \mathcal{Z}_{(\alpha_1, \alpha_2)} = \mathcal{Y} - \alpha_1 - \alpha_2$ where $\mathcal{Y}$ denotes the label set, which is identical for all anchor pairs. In $\tilde{f}_{\text{add}}$, $\phi_\alpha^{\boldsymbol{X}}$ are distinct across anchors $\alpha$ while $\phi_\alpha^y$ are identical for all $\alpha \in \mathcal{A}$.

- **Module addition** (Both signatures identical). $y = f_{\text{mod}}\left(\boldsymbol{X}\right) = \min\mathcal{Z} + (z + \alpha_1 + \alpha_2 \mod |\mathcal{Z}|), \quad \alpha_1, \alpha_2 \in \mathcal{A}$ and $z \in \mathcal{Z}$. Both $\phi_\alpha^{\boldsymbol{X}}$ and $\phi_\alpha^y$ are identical with different anchors.

In this work, we set $\mathcal{A} = \{11, 12, \cdots, 20\}$ and $\mathcal{Y} = \mathcal{Z} = \{101, 102, \cdots, 140\}$. Figure 2A visualizes the probability signature similarities for each task, confirming our manipulations. The detailed mathematical formulations of these signatures in each task are provided in Appendix B.1.

**Results: Theory Predicts Embedding Structure** We train $F_{\text{lin}}$ for each task with $d = 200$. Tasks $f_{\text{add}}$ and $\tilde{f}_{\text{add}}$ are well learned, while $f_{\text{mod}}$ fails to be fitted. The details are provided in Appendix A. Figure 2 B represents the value of $\cos\left(\boldsymbol{W}_\alpha^E, \boldsymbol{W}_{\alpha'}^E\right)$ in the three tasks.

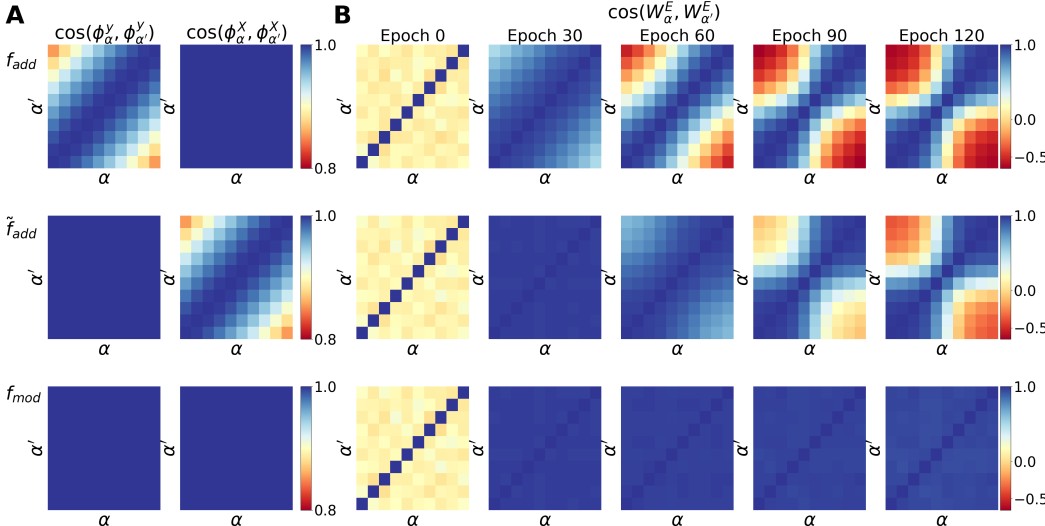

Figure 2: A: The heatmap of $\cos\left(\phi_\alpha^y, \phi_{\alpha'}^y\right)$ and $\cos\left(\phi_\alpha^X, \phi_{\alpha'}^X\right)$ in three addition tasks. B: The heatmap of $\cos\left(W_\alpha^E, W_{\alpha'}^E\right)$ in $F_{\mathrm{lin}}$ across different tasks.

- **Task $f_{\mathrm{add}}$:** different anchor embeddings quickly form an ordered structure, where the cosine similarity gets smaller as the anchor distance gets larger. The distribution of $\cos\left(W_\alpha^E, W_{\alpha'}^E\right)$ is consistent with the $\cos\left(\phi_\alpha^y, \phi_{\alpha'}^y\right)$ (Figure 2 A), implying the impact of $\phi_\alpha^y$ in directing $W_\alpha^E$.

- **Task $\tilde{f}_{\mathrm{add}}$:** The anchor embeddings also develop a similar hierarchical structure, aligned with the structure of $\phi_\alpha^X$ in $\tilde{f}_{\mathrm{add}}$. But its convergence is slower, validating that $\phi_\alpha^y$ dominates early dynamics.

- **Task $f_{\mathrm{mod}}$:** Although the task is unsolvable by a linear model, all anchor embeddings collapse to the same direction, exactly as Corollary 1 predicts when both signatures are identical.

### 4.2 FFN Unlocks Joint relationships: Solving the Modular Addition Puzzle

Recall that in Section 4.1, the linear model failed to learn $f_{\mathrm{mod}}$, whose embeddings collapsed to a single direction. It's not because the task lacked structure, but the linear model cannot encode the probability signature $\phi_x^{X|y}$. We find that the nonlinear activation could resolve this problem and provide the following results.

**Corollary 2** (Embedding of FFN). *Let $N \to \infty$, $\pi$ denotes the data distribution over the training set. The gradient flow of $W_x^E$ in $F_{\mathrm{ffn}}$ could be approximated by*

$$\frac{dW_x^E}{dt} = r_x^{\mathrm{in}}\left(W^{U,T}\left(\phi_x^y - \frac{1}{d_{\mathrm{vob}}}W^U W^E \phi_x^X\right) + \mathbb{T} \cdot \boldsymbol{\phi_x^{X|y}} + \boldsymbol{\epsilon}\right), \tag{2}$$

*where $\mathbb{T} \in \mathbb{R}^{d \times d_{\mathrm{vob}} \times d_{\mathrm{vob}}}$, $\mathbb{T}_{:,x',\nu} = W_\nu^U \odot W_{x'}^E$ for $\nu, x' \in \mathcal{V}$ and 0 otherwise. $\boldsymbol{\epsilon}$ represents the higher-order term.*

This is a qualitative leap beyond $F_{\mathrm{lin}}$: The new term $\mathbb{T} \cdot \phi_x^{X|y}$ directly encodes how the presence of $x$ influences the co-occurrence distribution conditioned on future labels. For $f_{\mathrm{mod}}$, $\phi_x^{X|y}$ varies systematically with $\alpha$ (shown in Figure 3 A), thereby providing the necessary signal that the linear model could not access. We train the $f_{\mathrm{mod}}$ with $F_{\mathrm{ffn}}$ to test whether $\phi_x^{X|y}$ enables structure formation. Figure 3 B depicts the cosine similarity among anchor embeddings, demonstrating that the embedding structure in $f_{\mathrm{mod}}$ is ordered, which validates our analysis. This contract validates that the specific probability signatures encoded are architecture-dependent, but the governing principle—gradient flow transforms signatures into structure—is universal.

**Geometric Proof: PCA Visualization of Signature-Embedding Alignment** Proposition 1 and Corollaries 1-2 make algebraic predictions; we now render them as visible geometry. Figure 4

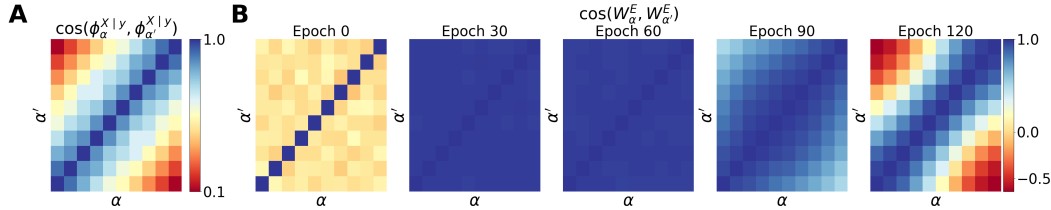

Figure 3: A: The heatmap of $\cos\left(\phi_\alpha^{\boldsymbol{X}|y}, \phi_{\alpha'}^{\boldsymbol{X}|y}\right)$ in $f_{\mathrm{mod}}$. B: $\cos\left(\boldsymbol{W}_\alpha^E, \boldsymbol{W}_{\alpha'}^E\right)$ in $F_{\mathrm{ffn}}$ learning $f_{\mathrm{mod}}$.

projects all probability signatures (left 3 columns) and learned embeddings (right 2 columns) into 2D space via PCA. This result reveals that in $F_{\mathrm{lin}}$, the embedding structure is primarily influenced by $\phi_\alpha^y$ and $\phi_\alpha^{\boldsymbol{X}}$. Specifically, when both $\phi_\alpha^y$ and $\phi_\alpha^{\boldsymbol{X}}$ are controlled in $f_{\mathrm{mod}}$, the embedding structure is chaotic. Besides, the embedding space in $F_{\mathrm{ffn}}$ is impacted by another probability signature $\phi_\alpha^{\boldsymbol{X}|y}$. These phenomena are consistent with our theoretical analysis, illustrating that analyzing the embedding space via the gradient flow and linking to the token relationships is viable.

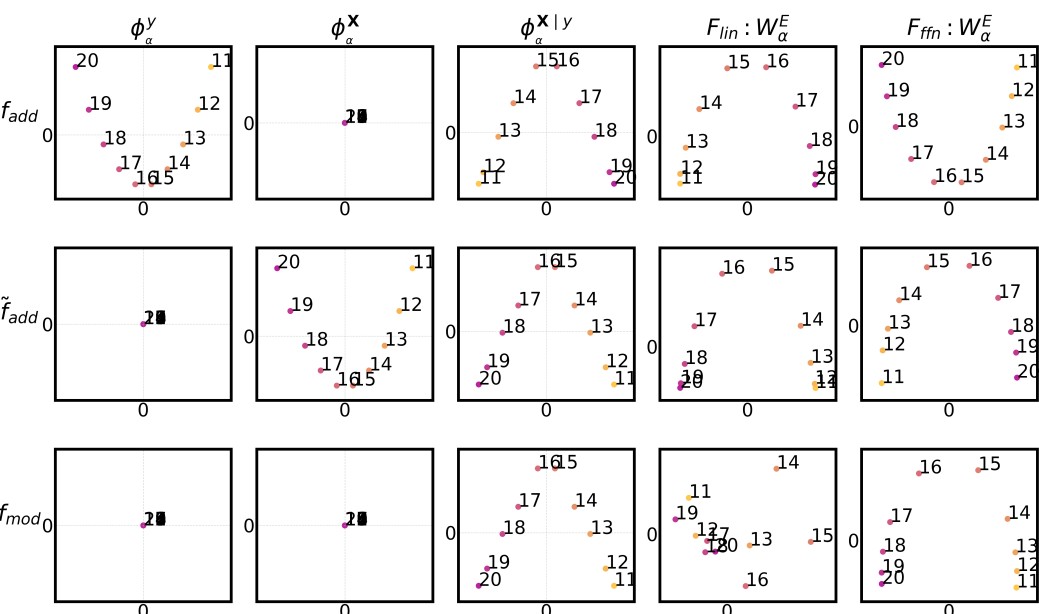

Figure 4: PCA projection of the three types of probability signatures and the embedding vectors in $F_{\mathrm{lin}}$ and $F_{\mathrm{ffn}}$ (epoch 120).

## 5 GRADIENT FLOW OF UNEMBEDDING VECTOR

Our analysis thus far has focused on the encoding side—how tokens are embedded into hidden space. A complete theory must also explain the decoding side: how the unembedding matrix $\boldsymbol{W}^U$ learns to map hidden representations back to token probabilities. Remarkably, gradient flow reveals a perfect symmetry: just as embeddings evolve under token-level probability signatures, unembeddings evolve under inverse signatures that capture how tokens are predicted from contexts.

**Proposition 2.** *Given an embedding-based model $F$ with an unembedding matrix $\boldsymbol{W}^U$. For any token $\nu \in \mathcal{V}$, the gradient flow of $\boldsymbol{W}_\nu^U$ (the $\nu$-th row of $\boldsymbol{W}^U$) can be written as*

$$\frac{d\boldsymbol{W}_\nu^U}{dt} = r_\nu^{\mathrm{out}}\mathbb{E}_\pi\left[G\left(\boldsymbol{W}_{\boldsymbol{X}}^E\right)^T \mid y = \nu\right] - \mathbb{E}_\pi\left[\boldsymbol{p}_\nu G\left(\boldsymbol{W}_{\boldsymbol{X}}^E\right)^T\right],$$

*where $r_\nu^{\mathrm{out}}$ denotes the ratio of sequences whose label is $\nu$ and $\boldsymbol{p}_\nu$ means the $\nu$-th element of $\boldsymbol{p}$.*

Specifically, we have the following formulation for the linear model:

**Corollary 3** (Unembedding of Linear Model). *Let $N \to \infty$, $\pi$ denotes the data distribution over the training set. The gradient flow of $\boldsymbol{W}_\nu^U$ in $F_{\text{lin}}$ could be approximated by*

$$\frac{d\boldsymbol{W}_\nu^U}{dt} = Lr_\nu^{\text{out}} \left(\boldsymbol{W}^E \boldsymbol{\varphi}_\nu^{\boldsymbol{X}}\right)^T + \boldsymbol{\eta}, \tag{3}$$

*where $\boldsymbol{\eta}$ denotes the output term.*

Corollary 3 demonstrates that $\phi_\nu^{\boldsymbol{X}}$ directs the dynamics of the unembedding vector. We extract the unembedding matrix from the addition tasks and compare its geometry to $\varphi_\nu^{\boldsymbol{X}}$. Figure 5 reveals the same striking alignment observed for embeddings. Figure 5 B depicts the distribution of $\cos\left(\varphi_\nu^{\boldsymbol{X}}, \varphi_{\nu'}^{\boldsymbol{X}}\right)$, which is aligned with the distribution of the $\cos\left(\boldsymbol{W}_\nu^U, \boldsymbol{W}_{\nu'}^U\right)$. Furthermore, Figure 5 C compares the PCA projection of $\varphi_\nu^{\boldsymbol{X}}$ and $\boldsymbol{W}_\nu^U$ in all tasks, revealing a high consistency and validating our analysis. This symmetric validation completes our framework: Gradient flow does not arbitrarily shape parameters—it encodes data statistics into model weights with mathematical precision, whether on the input or output side.

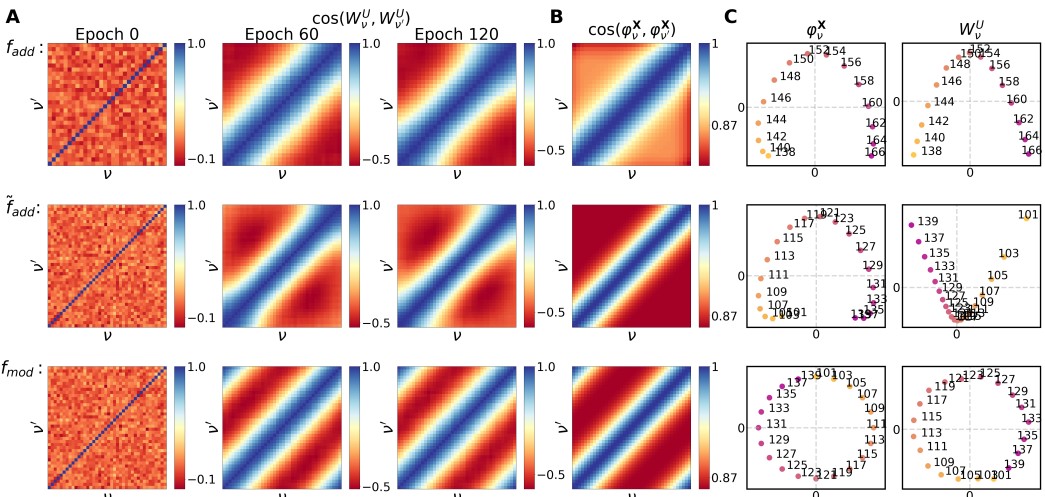

Figure 5: A: The heatmap of the $\cos\left(\boldsymbol{W}_\nu^U, \boldsymbol{W}_{\nu'}^U\right)$ in $F_{\text{lin}}$ during the training process. B: The heatmap of $\cos\left(\varphi_\nu^{\boldsymbol{X}}, \varphi_{\nu'}^{\boldsymbol{X}}\right)$ across different tasks. C: PCA projection of $\varphi_\nu^{\boldsymbol{X}}$ and $\boldsymbol{W}_\nu^U$ (epoch 120).

# 6 LANGUAGE MODEL

Our analysis of synthetic tasks demonstrates that gradient flow dynamics encode probability signatures into embedding structures. We now ask: Does this principle scale to language models trained on real-world corpora? A full analysis of all terms in Proposition 1 for Transformers would be intractable and, more importantly, unnecessary for validating our core contribution. We therefore adopt a minimalist validation strategy: analyze the dominant probability signature predicted by gradient flow and test whether it alone can predict embedding structure. If this simplified analysis succeeds, it proves that our framework captures the essential mechanism and researchers can then extend it to additional modules as needed.

For decoder-only Transformers with next-token prediction, the gradient flow of embeddings is dominated by the next-token distribution since the model could be formulated as follows.

$$F_{\text{lan}}\left(\boldsymbol{X}\right) = \boldsymbol{W}^U \left(\boldsymbol{W}_{\boldsymbol{X}}^E + \tilde{F}\left(\boldsymbol{X}\right)\right).$$

Formally, given the training corpus $\left\{\boldsymbol{X}^i\right\}_{i=1}^N$, we define the following probability signatures for any $s \in \mathcal{V}$:

$$\phi_s^{\text{next}} = \sum_{s' \in \mathcal{V}} \mathbb{P}_\pi \left(\cup_{t=1}^{L-1} \{X_{t+1} = s' \mid X_t = s\}\right) \boldsymbol{e}_{s'},$$

$$\varphi_s^{\text{pre}} = \sum_{s' \in \mathcal{V}} \mathbb{P}_\pi \left(\cup_{t=1}^{L-1} \{X_t = s' \mid X_{t+1} = s\}\right) \boldsymbol{e}_{s'}, \tag{4}$$

We derive the following result:

**Corollary 4.** *Let $N \to \infty$, $\pi$ denotes the token distribution in the training dataset. The gradient flow of the embedding vector $\boldsymbol{W}_s^E$ of token $s$ could be fomulated as*

$$\frac{d\boldsymbol{W}_s^E}{dt} = r_s^{\text{in}} \boldsymbol{W}^{U,T} \boldsymbol{\phi}_s^{\text{next}} + \boldsymbol{\eta}^E.$$

*Furthermore, the gradient flow of the unembedding vector $\boldsymbol{W}_s^U$ could be approximated as*

$$\frac{d\boldsymbol{W}_s^U}{dt} = r_s^{\text{out}} \left( \boldsymbol{W}^E \boldsymbol{\varphi}_s^{\text{pre}} \right)^T + \boldsymbol{\eta}^U.$$

*The $\boldsymbol{\eta}^E$ and $\boldsymbol{\eta}^U$ denote the output probability and the higher-order term.*

**Probability signatures impact the embedding space in language models** Corollary 4 suggests that given any token $s$, the distributions of its next token and previous token significantly impact its embedding. To verify this result, we trained a group of Qwen2.5 models on different subsets of the Pile. Figure 6 A shows these similarity matrices for the dataset Pile-dm-mathematics, where the tokens displayed are those that occur most frequently in the corpus. We define the following correlation coefficient $R_{\cos} \left( \boldsymbol{W}^E, \boldsymbol{\phi}^{\text{next}} \right) := \text{Corr} \left( \cos \left( \boldsymbol{W}_s^E, \boldsymbol{W}_{s'}^E \right), \cos \left( \boldsymbol{\phi}_s^{\text{next}}, \boldsymbol{\phi}_{s'}^{\text{next}} \right) \right)$, and similarly $R_{\cos} \left( \boldsymbol{W}^U, \boldsymbol{\varphi}^{\text{pre}} \right)$. Figure 6 B tracks the $R_{\cos} \left( \boldsymbol{W}^E, \boldsymbol{\phi}^{\text{next}} \right)$ and $R_{\cos} \left( \boldsymbol{W}^U, \boldsymbol{\varphi}^{\text{pre}} \right)$ across all subsets during training (20 epochs). Correlations increase during the first epoch, indicating that gradient flow rapidly encodes next-token and previous-token statistics into embeddings and unembeddings. After reaching peak alignment, correlations plateau and dip slightly, showing that the embedding structure is still largely impacted by $\boldsymbol{\phi}_s^{\text{next}}$ and $\boldsymbol{\varphi}_s^{\text{pre}}$. The fact that a single simplified probability signature maintains predictive power throughout training, proves that our gradient flow analysis captures the essential mechanism of embedding structure. Researchers can now systematically uncover additional probability signatures (e.g., from attention patterns or higher-order terms) to account for residual variance. Furthermore, we find that the probability signatures reflect the strong connections of embeddings more faithfully, and we provide a detailed analysis in the Appendix C.3. Additionally, we provide another set of experiments using the Llama2 architecture in Appendix C.4.

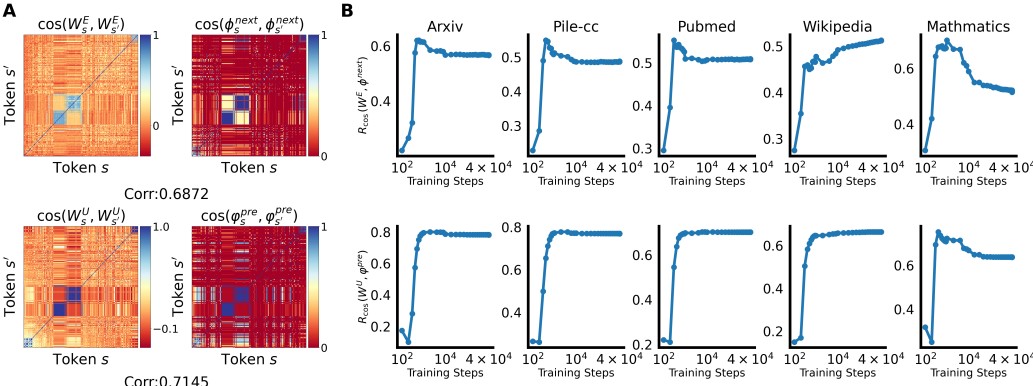

Figure 6: A: Heatmap of $\cos \left( \boldsymbol{W}_s^E, \boldsymbol{W}_{s'}^E \right)$ (left up), $\cos \left( \boldsymbol{\phi}_s^{\text{next}}, \boldsymbol{\phi}_{s'}^{\text{next}} \right)$ (right up), $\cos \left( \boldsymbol{W}_s^U, \boldsymbol{W}_{s'}^U \right)$ (left down) and $\cos \left( \boldsymbol{\varphi}_s^{\text{pre}}, \boldsymbol{\varphi}_{s'}^{\text{pre}} \right)$ (right up) in the experiment on dataset Pile-dm-mathematics (1 epoch). B: The dynamics of $R_{\cos} \left( \boldsymbol{W}^E, \boldsymbol{\phi}^{\text{next}} \right)$ (top) and $R_{\cos} \left( \boldsymbol{W}^U, \boldsymbol{\varphi}^{\text{pre}} \right)$ (bottom) during training (20 epochs) across different datasets.

**Validating with the open-source model** Since general-purpose pretrained base models are trained on broad corpora, we attempt to directly estimate their embedding structure by the probability signature. We employ Qwen2.5-3B-base for comparison and define $\tilde{\boldsymbol{\phi}}_s = \boldsymbol{\phi}_s^{\text{next}} + \boldsymbol{\varphi}_s^{\text{pre}}$, since $\boldsymbol{W}^E = \boldsymbol{W}^{U,T}$ in Qwen2.5-3B-base (the detail is provided in Appendix C.2). We compute $\tilde{\boldsymbol{\phi}}_s$ from the subsets of Pile. As shown in Figure 7 A, the structure of $\tilde{\boldsymbol{\phi}}_s$ could capture the main properties of the embedding structure, particularly the presence of sub-blocks with high similarity. Furthermore, we examine the instance for the digits ranging from 1 to 9. Figure 1 exhibits the PCA projections of $\boldsymbol{W}_s^E$ and $\tilde{\boldsymbol{\phi}}_s$, while Figure 7B illustrates their respective cosine similarities $\cos \left( \boldsymbol{W}_s^E, \boldsymbol{W}_{s'}^E \right)$ and

$\cos\left(\tilde{\phi}_s, \tilde{\phi}_{s'}\right)$, with both figures revealing an ordered organization aligned with the numerical sequence. However, this estimation does not always hold. On the one hand, Zhang et al. (2024) finds that initialization scale significantly affects the emergence of such embedding structures, demonstrating that in the NTK regime, the embedding structure may fail to capture token relationships. On the other hand, since probability signatures are computed from the training dataset, obtaining the correct data distribution becomes difficult when the corpus is carefully curated.

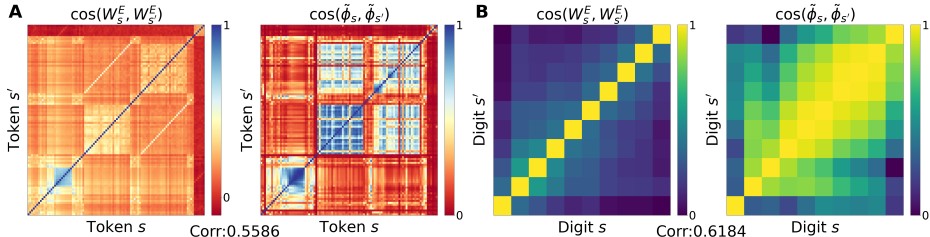

Figure 7: $\cos\left(\boldsymbol{W}_s^E, \boldsymbol{W}_{s'}^E\right)$ of the Qwen2.5-3B-base and $\cos\left(\tilde{\phi}_s, \tilde{\phi}_{s'}\right)$, respectively, with the frequently-appearing tokens (A) and the digits from 0 to 9 (B).

## 7 DISCUSSION & CONCLUSION

We have shown that the geometry of embedding spaces is not a mysterious emergent phenomenon, but a deterministic encoding of probability signatures sculpted by gradient flow dynamics. More importantly, we have demonstrated that this encoding can be reverse-engineered: given any embedding-based architecture, our framework systematically extracts the exact set of statistical relationships that drive embedding evolution. This transforms representation learning from a black box into a transparent, distribution-driven process.

**Guidance for Model Architectures and Training Methods** We illustrate that each architecture implicitly selects which probability signatures it can encode. Our gradient-flow analysis makes this selection explicit and quantifiable: Corollary 1 proves that linear models cannot encode joint token-label relationships ($\phi_x^{\boldsymbol{X}|y}$). Any task requiring this relationship will fail, regardless of scale. Adding a nonlinear activation unlocks $\phi_x^{\boldsymbol{X}|y}$ (Corollary 2), enabling models to learn such semantics. This suggests a principled architecture search: introduce modules whose Jacobians $G^{(1)}$ encode desired probability signatures. On the other hand, our results have shown that the loss function is not merely a performance metric but also a gradient flow sculptor that determines which probability signatures dominate. Corollary 4 shows that next-token prediction makes $\phi_s^{\text{next}}$ the dominant signature, embedding tokens based on immediate neighbors. This explains why standard autoregressive models excel at local coherence but struggle with long-range dependencies. If the loss predicts $k$ future tokens, gradient flow will encode the k-gram relationship distribution. This provides a theoretical explanation for why multi-token prediction could easily capture the global relationships (Gloeckle et al., 2024).

**Future Work** We deliberately analyzed only four signature families and a simplified LLM gradient flow. This was not due to theoretical incompleteness, but to demonstrate the framework's modular extensibility. Just as we derived $\phi_x^{\boldsymbol{X}|y}$ for feedforward networks and $\phi_s^{\text{next}}$ for Transformers, researchers can now systematically mine custom signatures for their architectures of interest. The framework is designed to be extended. As a future direction, we will focus on analyzing the probability signatures in the self-attention module and the completed Transformer layer. This is not a correction to our theory, but its natural evolution.

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

## LLMs USAGE

In this work, the LLMs are employed to correct grammatical errors and inappropriate words.

## A EXPERIMENTAL SETUPS

**Addition tasks** For each type of addition task, we trained a linear model $F_{\text{lin}}$ and a Feedforward network $F_{\text{ffn}}$. The hidden size $d = 200$, and we employed the ReLU as the activation function. Each dataset contains 50000 data pairs. The training is conducted for 1000 epochs with a batch size of 100. The AdamW optimizer is employed with an initial learning rate of $10^{-5}$. Inspired by the work of Luo et al. (2021); Xu et al. (2025b), we initialize the model parameters by $\boldsymbol{W}_{i,j} \sim \mathcal{N}\left(0, d^{-0.8}\right)$, indicating a small initialization scale.

**Language models** In the analysis of the LLMs, we employ the Qwen2.5 architecture with 12 layers and 12 attention heads in each layer. We set up that the hidden size is 512, and the intermediate size in FFN is 1024. The dimension of the key vectors and value vectors in each head is 64. Similarly, we initialize the parameter by $\boldsymbol{W}_{i,j} \sim \mathcal{N}\left(0, d_{\text{in}}^{-1}\right)$ where $d_{\text{in}}$ means the input dimension of $\boldsymbol{W}$. We select five subsets of Pile, including Pile-arxiv, Pile-dm-mathematics, Pile-cc, Pile-pubmed-central, and Pile-wikipedia-en. The length of each sequence is 2048. The training is conducted for 1 epoch in each experiment, with the AdamW optimizer and a cosine learning rate schedule utilized. The initial learning rate is $10^{-4}$.

## B ADDITION TASK

### B.1 PROBABILITY SIGNATURES IN ADDITION TASKS

We provide a formulation of the following probability in the three addition tasks. We denote $U\left(\mathcal{A}\right)$ and $U\left(\mathcal{Z}\right)$ as the discrete uniform distribution over $\mathcal{A}$ and $\mathcal{Z}$, respectively. $A$ and $Z$ are the random variables following $U\left(\mathcal{A}\right)$ and $U\left(\mathcal{Z}\right)$. For the task $f_{\text{add}}$, we have that

$$\mathbb{P}_\pi\left(y = \nu \mid \alpha \in \boldsymbol{X}\right) = \mathbb{P}_\pi\left(A + Z = \nu - \alpha\right), \quad \mathbb{P}_\pi\left(z \in \mathcal{X} \mid \alpha \in \boldsymbol{X}\right) = \frac{1}{|\mathcal{Z}|},$$

$$\mathbb{P}_\pi\left(z \in \boldsymbol{X} \mid \alpha \in \boldsymbol{X}, y = \nu\right) = \mathbb{P}_\pi\left(A = \nu - \alpha - z\right) = \frac{1}{|\mathcal{A}|}\delta_{\nu - \alpha - z \in \mathcal{A}},$$

$$\mathbb{P}_\pi\left(\alpha' \in \boldsymbol{X} \mid \alpha \in \boldsymbol{X}, y = \nu\right) = \mathbb{P}_\pi\left(Z = \nu - \alpha - \alpha'\right) = \frac{1}{|\mathcal{Z}|}\delta_{\nu - \alpha - \alpha' \in \mathcal{Z}},$$

$$\mathbb{P}_\pi\left(z \in \boldsymbol{X} \mid y = \nu\right) = \mathbb{P}_\pi\left(A + A = \nu - z\right), \quad \mathbb{P}_\pi\left(\alpha \in \boldsymbol{X} \mid y = \nu\right) = \mathbb{P}_\pi\left(A + Z = \nu - \alpha\right),$$

where $\alpha, \alpha' \in \mathcal{A}, z \in \mathcal{Z}$. It's noted that besides the co-occurrence probability $\mathbb{P}_\pi\left(z \in \mathcal{X} \mid \alpha \in \boldsymbol{X}\right)$, the value of other ones is dependent on $\alpha$ or $\nu$. Figure 8 (left) displays the distribution of these probabilities, which intuitively reveals the cause of the hierarchy structure in the similarity matrix. Similarly, for $\tilde{f}_{\text{add}}$, denote $Y \sim U\left(\mathcal{Y}\right)$ and we have

$$\mathbb{P}_\pi\left(y = \nu \mid \alpha \in \boldsymbol{X}\right) = \frac{1}{|\mathcal{Y}|}, \quad \mathbb{P}_\pi\left(z \in \mathcal{X} \mid \alpha \in \boldsymbol{X}\right) = \mathbb{P}_\pi\left(Y - A = z + \alpha\right),$$

$$\mathbb{P}_\pi\left(z \in \boldsymbol{X} \mid \alpha \in \boldsymbol{X}, y = \nu\right) = \mathbb{P}_\pi\left(A = \nu - \alpha - z\right) = \frac{1}{|\mathcal{A}|}\delta_{\nu - \alpha - z \in \mathcal{A}},$$

$$\mathbb{P}_\pi\left(\alpha' \in \boldsymbol{X} \mid \alpha \in \boldsymbol{X}, y = \nu\right) = \frac{1}{|\mathcal{Z}|},$$

$$\mathbb{P}_\pi\left(z \in \boldsymbol{X} \mid y = \nu\right) = \mathbb{P}_\pi\left(A + A = \nu - z\right), \quad \mathbb{P}_\pi\left(\alpha \in \boldsymbol{X} \mid y = \nu\right) = \mathbb{P}_\pi\left(A + Z = \nu - \alpha\right).$$

For $f_{\mathrm{mod}}$, we have

$$\mathbb{P}_\pi\left(y = \nu \mid \alpha \in \boldsymbol{X}\right) = \frac{1}{|\mathcal{Z}|}, \quad \mathbb{P}_\pi\left(z \in \mathcal{X} \mid \alpha \in \boldsymbol{X}\right) = \frac{1}{|\mathcal{Z}|},$$

$$\mathbb{P}_\pi\left(z \in \boldsymbol{X} \mid \alpha \in \boldsymbol{X}, y = \nu\right) = \frac{1}{|\mathcal{A}|}\delta_{\nu - \min \mathcal{Z} - (\alpha - z \bmod |\mathcal{Z}|) \in (A \bmod |\mathcal{Z}|)},$$

$$\mathbb{P}_\pi\left(\alpha' \in \boldsymbol{X} \mid \alpha \in \boldsymbol{X}, y = \nu\right) = \frac{1}{|\mathcal{Z}|},$$

$$\mathbb{P}_\pi\left(z \in \boldsymbol{X} \mid y = \nu\right) = \mathbb{P}_\pi\left((A + A \bmod |\mathcal{Z}|) = \nu - \min \mathcal{Z} - (z \bmod |\mathcal{Z}|)\right),$$

$$\mathbb{P}_\pi\left(\alpha \in \boldsymbol{X} \mid y = \nu\right) = \mathbb{P}_\pi\left((A + Z \bmod |\mathcal{Z}|) = \nu - \min \mathcal{Z} - (\alpha \bmod |\mathcal{Z}|)\right).$$

Figure 8 depicts all these probability distributions.

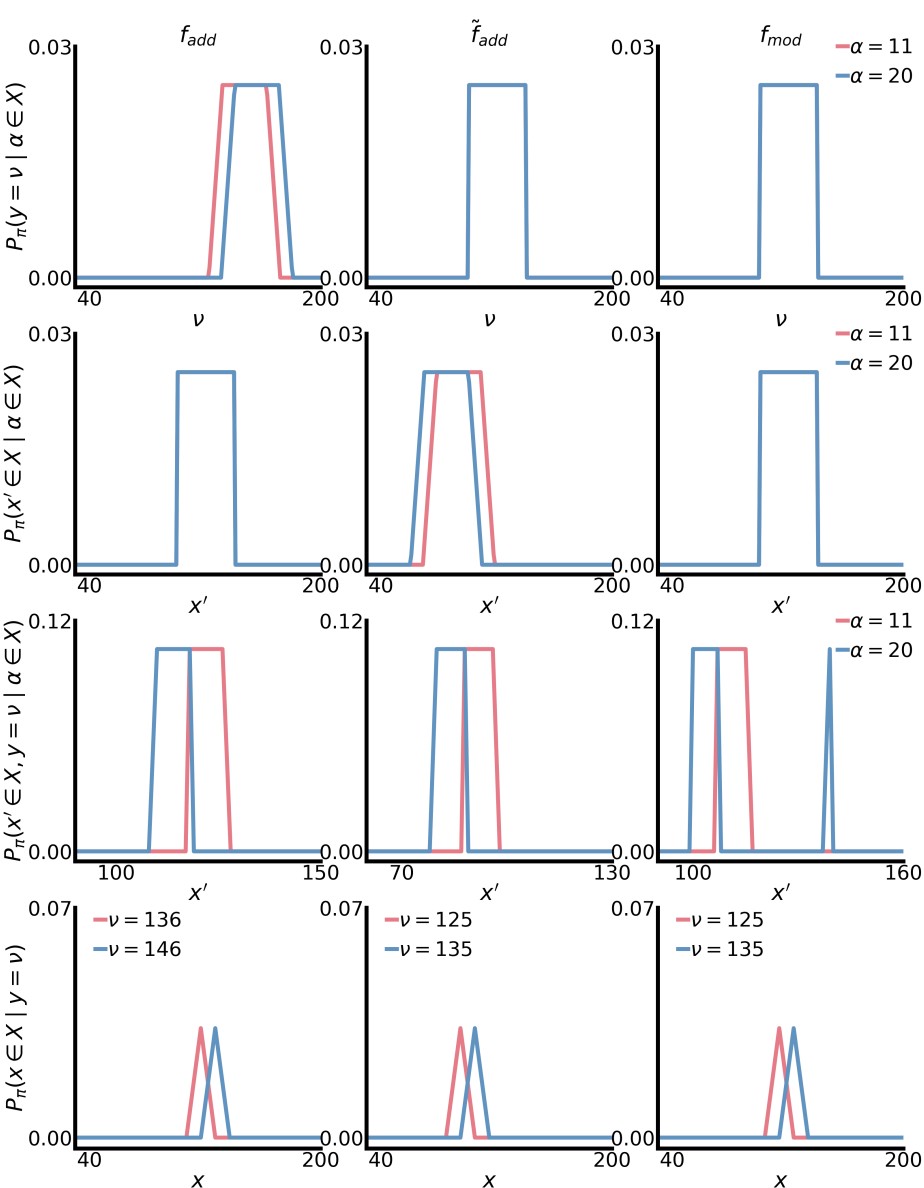

Figure 8: Probability signatures in each task under distinct $\alpha$ and $\nu$. In the distribution of $\mathbb{P}_\pi\left(x' \in \boldsymbol{X}, y = \nu \mid \alpha \in \boldsymbol{X}\right)$, $\nu = 150$ is displayed in $f_{\text{add}}$ and $\nu = 120$ in $\tilde{f}_{\text{add}}$ and $f_{\text{mod}}$, since 150 and 120 are the average label value in each task.

## B.2 TRAINING RESULT

Figure 9 shows the training accuracy of $F_{\text{lin}}$ and $F_{\text{ffn}}$ on the three addition tasks. The results reveal that both $f_{\text{add}}$ and $\tilde{f}_{\text{add}}$ are learned well by the linear model, whereas $f_{\text{mod}}$ requires the nonlinear model to achieve an effective fit.

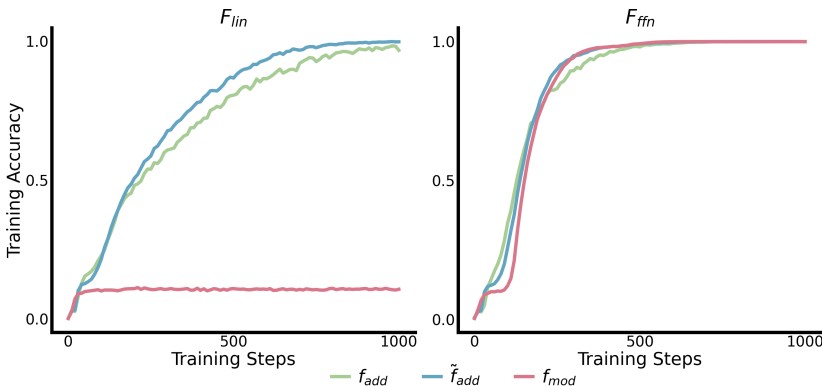

Figure 9: Training accuracy of the $F_{\text{lin}}$ (left) and $F_{\text{ffn}}$ (right) on the three addition tasks.

## B.3 QUANTIFY THE HIERARCHY EMBEDDING STRUCTURE

In the addition tasks, the anchors exhibit a strict ordering due to the numerical sequence. This provides an ideal setting for the embedding space to develop a corresponding ordered relationship. To formally quantify the formation of the ordered structure, we define the following metric:

$$R_{\text{order}}\left(\boldsymbol{W}_{\mathcal{A}}^{E}\right) = \text{Corr}\left(\cos\left(\boldsymbol{W}_{\alpha}^{E}, \boldsymbol{W}_{\alpha'}^{E}\right), |\,\alpha - \alpha'\,|\right).$$

$R_{\text{order}}\left(\boldsymbol{W}_{\mathcal{A}}^{E}\right)$ reflects the relationship between embedding similarity and anchor difference. A strong negative $R_{\text{order}}\left(\boldsymbol{W}_{\mathcal{A}}^{E}\right)$ (approximately $-1$) indicates that the similarity decreases systematically with increasing anchor difference, confirming the presence of a hierarchical organization in the anchor embeddings. Figure 10 depicts the corresponding evolution of $R_{\text{order}}\left(\boldsymbol{W}_{\mathcal{A}}^{E}\right)$ in $F_{\text{lin}}$ and $F_{\text{ffn}}$, which is consistent with our analysis.

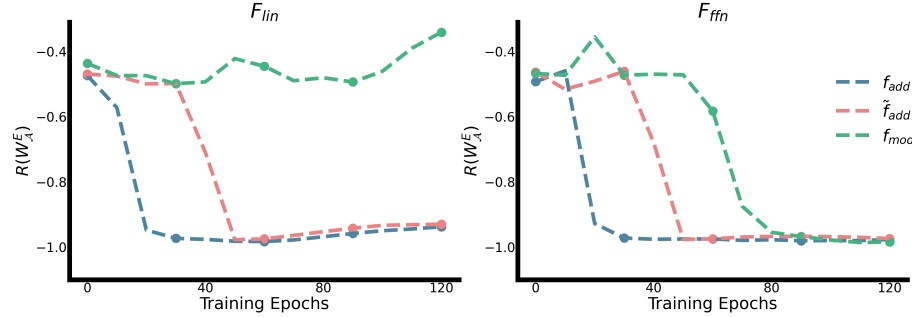

Figure 10: Dynamics of $R_{\text{order}}\left(\boldsymbol{W}_{\mathcal{A}}^{E}\right)$ in $F_{\text{lin}}$ (left) and $F_{\text{ffn}}$ (right). Line colors represent task types.

## B.4 UMEMBEDDING MATRIX IN FEEDFORWARD NETWORK

Figure 11 displays the structure of the unembedding matrix in $F_{\text{ffn}}$ with the three types of addition tasks. The distribution of $\cos\left(\boldsymbol{W}_{\nu}^{U}\right)$ (A) and the PCA projection (B) jointly reveal that the unembedding vectors of those label tokens establish a hierarchy structure, which is consistent with their natural sequence.

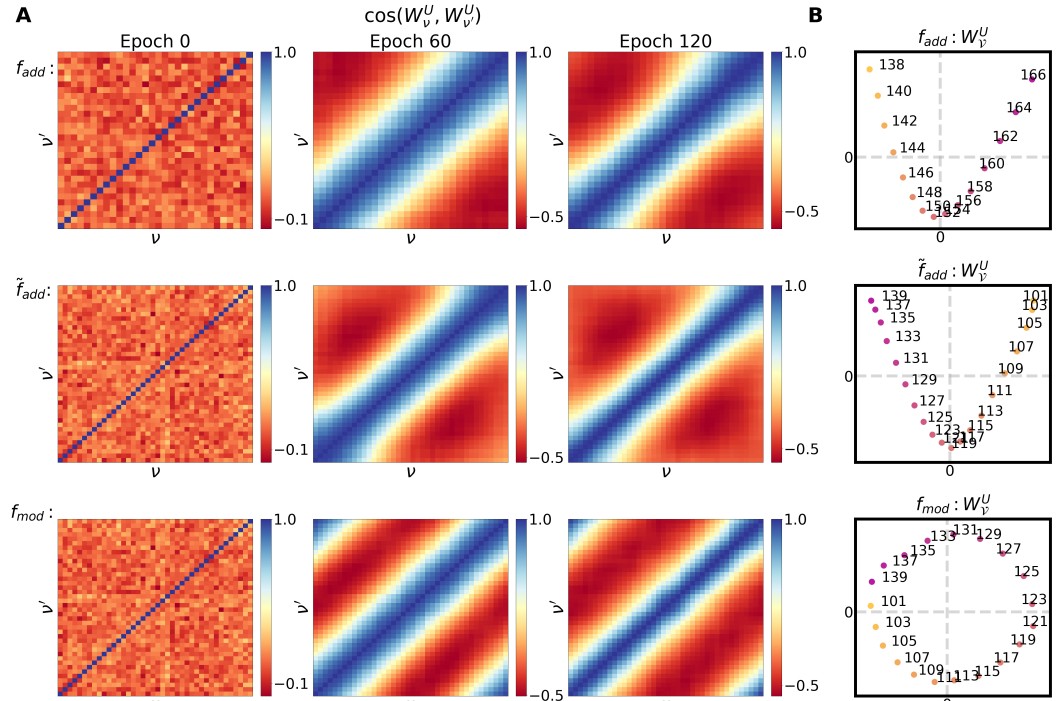

Figure 11: A: The heatmap of the $\cos\left(\boldsymbol{W}_{\mathcal{V}}^U\right)$ with label index in $F_{\text{ffn}}$ during the training process. B: PCA projection of $\boldsymbol{W}_{\mathcal{V}}^U$ in $F_{\text{ffn}}$ (epoch 120).

## C  LANGUAGE MODELS

### C.1  COMPLETE RESULTS

Figure 12 represents the cosine similarity distribution of $\boldsymbol{W}^E, \phi^{\text{next}}, \boldsymbol{W}^U$ and $\boldsymbol{\varphi}^{\text{pre}}$ at epoch 1 in the other 4 subsets of Pile we selected, exhibiting an analogous phenomenon with the observation in Figure 6. The distribution representations $\phi^{\text{next}}$ and $\boldsymbol{\varphi}^{\text{pre}}$ could effectively capture the high similarity among embedding vectors and unembedding vectors, respectively. Figure 13 depicts the comparison at epoch 20.

### C.2  TIED EMBEDDING

In the Qwen2.5-3B-base model, $\boldsymbol{W}^E = \boldsymbol{W}^{U,T}$, which aims for computational source saving. Under this condition, we have that

$$\frac{d\boldsymbol{W}_s^E}{dt} = r_s^{\text{in}}\boldsymbol{W}^{U,T}\phi_s^{\text{next}} + r_s^{\text{out}}\boldsymbol{W}^E\boldsymbol{\varphi}_s^{\text{pre}} + \boldsymbol{\eta}$$

$$= \boldsymbol{W}^E\left(r_s^{\text{in}}\phi_s^{\text{next}} + r_s^{\text{out}}\boldsymbol{\varphi}_s^{\text{pre}}\right) + \boldsymbol{\eta}.$$

Since the next-token-prediction, each token will be an input and an output, except the last token in a sequence, resulting in $r_s^{\text{in}} \approx r_s^{\text{out}}$. Denote $r_s = r_s^{\text{in}}$ and $\tilde{\phi}_s = \phi_s^{\text{next}} + \boldsymbol{\varphi}_s^{\text{pre}}$, then we have

$$\frac{d\boldsymbol{W}_s^E}{dt} = r_s\boldsymbol{W}^E\tilde{\phi}_s + \boldsymbol{\eta}.$$

### C.3  PROBABILITY SIGNATURE CAPTURE STRONG EMBEDDING SIMILARITIES

We find that the probability signatures reflect the strong connections of embeddings more faithfully. As shown in Figure 14 A, the correlation between $\text{Corr}\left(\cos\left(\boldsymbol{W}_s^E, \boldsymbol{W}^E\right), \cos\left(\phi_s^{\text{next}}, \phi^{\text{next}}\right)\right)$ and

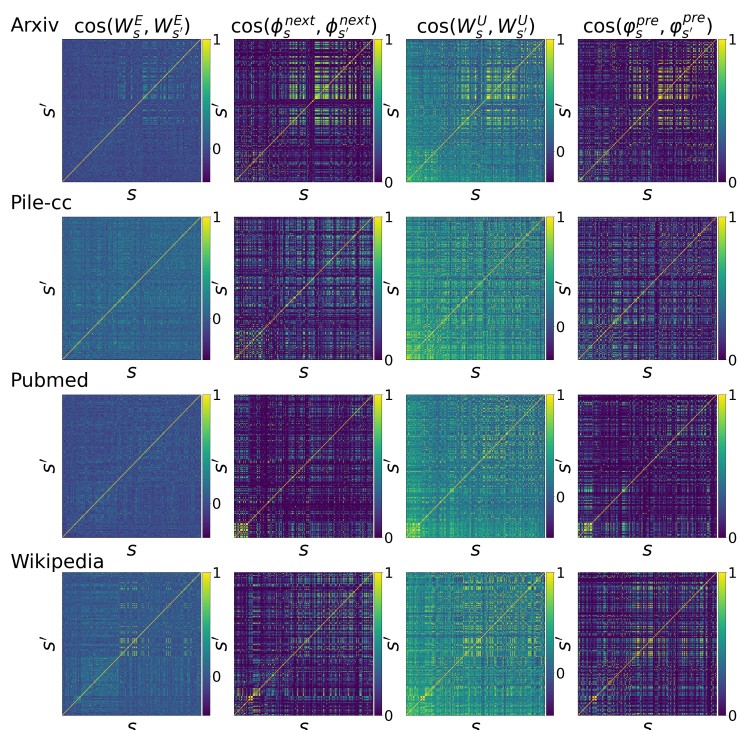

Figure 12: Heatmap of $\cos\left(\boldsymbol{W}_s^E, \boldsymbol{W}_{s'}^E\right)$ (left up), $\cos\left(\boldsymbol{\phi}_s^{\text{next}}, \boldsymbol{\phi}_{s'}^{\text{next}}\right)$ (right up), $\cos\left(\boldsymbol{W}_s^U, \boldsymbol{W}_{s'}^U\right)$ (left down) and $\cos\left(\boldsymbol{\varphi}_s^{\text{pre}}, \boldsymbol{\varphi}_{s'}^{\text{pre}}\right)$ (right up) (epoch 1) in each experiment with distinct dataset. The tokens displayed are those with the most appearances in the dataset.

$\cos\left(\boldsymbol{W}_s^E, \boldsymbol{W}^E\right)$ is plotted against for all tokens $s$, demonstrating stronger consistency in high-similarity regions. We define $p_{\cos(\boldsymbol{W}^E)}$ and $p_{\cos(\boldsymbol{\phi}^{\text{next}})}$ as the percentile matrix of each elements in $\cos\left(\boldsymbol{W}^E\right)$ and $\cos\left(\boldsymbol{\phi}^{\text{next}}\right)$, respectively. Figure 14 B displays the distribution of $p_{\cos(\boldsymbol{\phi}^{\text{next}})}$, conditioned on different intervals of the $p_{\cos(\boldsymbol{W}^E)}$, and Figure 14 C shows the average value of $p_{\cos(\boldsymbol{\phi}^{\text{next}})}$ within each interval of $p_{\cos(\boldsymbol{W}^E)}$. It can be observed that the alignment is significantly stronger in the regions with large embedding similarity.

**Remark about Figure 14 A**  In each subset $D_i, i = 1, 2, \cdots M$, we define the set $\mathcal{S}_i = \left\{s_j^i\right\}_{j=1}^{C_i}$ as the set of the $C_i$ tokens which appear most frequently in $D_i$. Based on the dataset $D_i$, and denote $\boldsymbol{W}^{E_i}$ as the embedding matrix of the model corresponding to dataset $D_i$, we compute that

$$\cos_{D_i}\left(\boldsymbol{W}_{s_j^i}^E, \boldsymbol{W}^E\right) = \left[\cos\left(\boldsymbol{W}_{s_j^i}^{E_i}, \boldsymbol{W}_{s'}^{E_i}\right)\right]_{s' \in \mathcal{S}_i} \in \mathbb{R}^{C_i},$$

and

$$\cos_{D_i}\left(\boldsymbol{\phi}_{s_j^i}^{\text{next}}, \boldsymbol{\phi}^{\text{next}}\right) = \left[\cos\left(\boldsymbol{\phi}_{s_j^i}^{\text{next}}, \boldsymbol{\phi}_{s'}^{\text{next}}\right)\right]_{s' \in \mathcal{S}_i} \in \mathbb{R}^{C_i}.$$

for any token $s_j^i \in \mathcal{S}_i$. Then we define the correlation coefficient

$$R_{D_i}\left(s_j^i\right) = \text{Corr}\left(\cos_{D_i}\left(\boldsymbol{W}_{s_j^i}^E, \boldsymbol{W}^E\right), \cos_{D_i}\left(\boldsymbol{\phi}_{s_j^i}^{\text{next}}, \boldsymbol{\phi}^{\text{next}}\right)\right)$$

and the average embedding similarity as

$$\text{Mean}_{\boldsymbol{W}^E, D_i}\left(s_j^i\right) = \frac{1}{C_i}\cos_{D_i}\left(\boldsymbol{W}_{s_j^i}^E, \boldsymbol{W}^E\right) \cdot \mathbf{1}.$$

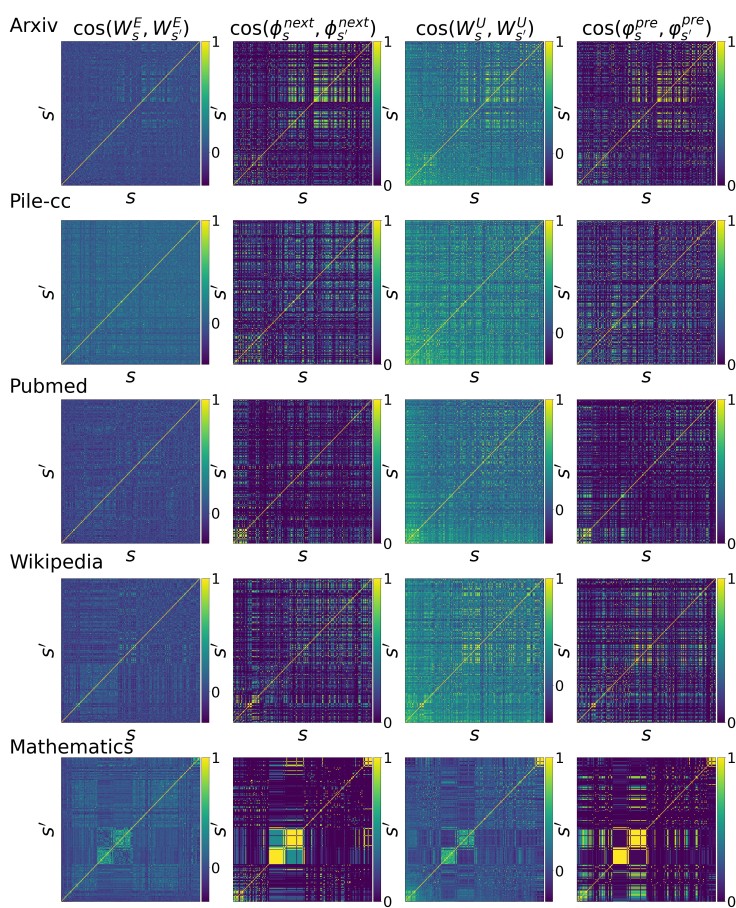

Figure 13: Heatmap of $\cos\left(\boldsymbol{W}_s^E, \boldsymbol{W}_{s'}^E\right)$ (left up), $\cos\left(\phi_s^{\text{next}}, \phi_{s'}^{\text{next}}\right)$ (right up), $\cos\left(\boldsymbol{W}_s^U, \boldsymbol{W}_{s'}^U\right)$ (left down) and $\cos\left(\boldsymbol{\varphi}_s^{\text{pre}}, \boldsymbol{\varphi}_{s'}^{\text{pre}}\right)$ (right up) (epoch 20) in each experiment with distinct dataset. The tokens displayed are those with the most appearances in the dataset.

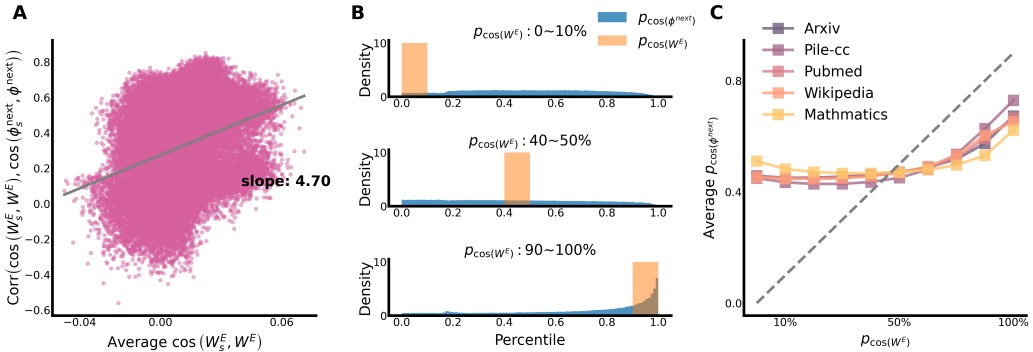

Figure 14: A: Relation between $\mathrm{Corr}\left(\cos\left(\boldsymbol{W}_s^E, \boldsymbol{W}^E\right), \cos\left(\phi_s^{\text{next}}, \phi^{\text{next}}\right)\right)$ and the average value of $\cos\left(\boldsymbol{W}_s^E, \boldsymbol{W}^E\right)$. Each point denotes a token $s$. B: Distribution of $p_{\cos(\phi^{\text{next}})}$, conditioned on intervals $0 \sim 10\%, 40 \sim 50\%$ and $90 \sim 100\%$ of the $p_{\cos(\boldsymbol{W}^E)}$. C: Average value of $p_{\cos(\phi^{\text{next}})}$ within each interval of $p_{\cos(\boldsymbol{W}^E)}$.

Then we concatenate the metrics with all token $s_j^i \in \mathcal{S}_i, j = 1, 2, \cdots, C_i$ and all datasets $\mathcal{S}_i, i = 1, 2, \cdots, M$, i.e.

$$\mathrm{Corr}\left(\cos\left(\boldsymbol{W}_s^E, \boldsymbol{W}^E\right), \cos\left(\boldsymbol{\phi}_s^{\mathrm{next}}, \boldsymbol{\phi}^{\mathrm{next}}\right)\right) = \left[R_{D_i}\left(s_j^i\right)\right]_{j=1,2,\cdots,C_i}^{i=1,2,\cdots,M} \in \mathbb{R}^{\sum_{i=1}^M C_i},$$

$$\mathrm{Mean}\left(\cos\left(\boldsymbol{W}_s^E, \boldsymbol{W}^E\right)\right) = \left[\mathrm{Mean}_{\boldsymbol{W}^E, D_i}\left(s_j^i\right)\right]_{j=1,2,\cdots,C_i}^{i=1,2,\cdots,M} \in \mathbb{R}^{\sum_{i=1}^M C_i}.$$

Figure 6 displays the relation between $\mathrm{Corr}\left(\cos\left(\boldsymbol{W}_s^E, \boldsymbol{W}^E\right), \cos\left(\boldsymbol{\phi}_s^{\mathrm{next}}, \boldsymbol{\phi}^{\mathrm{next}}\right)\right)$ and $\mathrm{Mean}\left(\cos\left(\boldsymbol{W}_s^E, \boldsymbol{W}^E\right)\right)$, revealing a positive correlation. In our work, $M = 5$, and we set up $C_i = 10000$ for each dataset.

**Remark about Figure 14 B & C** In each subset $D_i, i = 1, 2, \cdots M$, we define the set $\mathcal{S}_i = \left\{s_j^i\right\}_{j=1}^{C_i}$ as the set of the $C_i$ tokens which appear most frequently in $D_i$. We compute that

$$\cos_{D_i}\left(\boldsymbol{W}^E\right) = \left[\cos\left(\boldsymbol{W}_s^{E_i}, \boldsymbol{W}_{s'}^{E_i}\right)\right]_{s,s'\in\mathcal{S}_i} \in \mathbb{R}^{C_i \times C_i}$$

and

$$\cos_{D_i}\left(\boldsymbol{\phi}^{\mathrm{next}}\right) = \left[\cos\left(\boldsymbol{\phi}_s^{\mathrm{next}}, \boldsymbol{\phi}_{s'}^{\mathrm{next}}\right)\right]_{s,s'\in\mathcal{S}_i} \in \mathbb{R}^{C_i \times C_i}.$$

Then translate the similarity matrix into a percentile formulation, i.e.

$$p_{\cos_{D_i}(\boldsymbol{W}^E)} = \mathrm{Percentile}\left(\cos_{D_i}\left(\boldsymbol{W}^E\right)\right), \quad p_{\cos_{D_i}(\boldsymbol{\phi}^{\mathrm{next}})} = \mathrm{Percentile}\left(\cos_{D_i}\left(\boldsymbol{\phi}^{\mathrm{next}}\right)\right)$$

and $p_{\cos(\boldsymbol{W}^E)} = \left[p_{\cos_{D_i}(\boldsymbol{W}^E)}\right]_{i=1,2,\cdots,M}$, $p_{\cos(\boldsymbol{\phi}^{\mathrm{next}})} = \left[p_{\cos_{D_i}(\boldsymbol{\phi}^{\mathrm{next}})}\right]_{i=1,2,\cdots,M}$. Figure 6 D and E reveal the distribution and average value of $p_{\cos(\boldsymbol{\phi}^{\mathrm{next}})}$, where $k \times 10\% \leq p_{\cos(\boldsymbol{W}^E)} < (k+1) \times 10\%, k = 0, 1, 2, \cdots, 9$.

**Case Analysis** We provide a detailed case to explain the group of tokens exhibiting high embedding similarities. In experiments on the Pile-dm-mathematics dataset, tokens such as "/a", "/b", "/c", and "/d" often serve as denominators in mathematical expressions. Figure 15 shows the cosine similarities of both their embedding vectors and distribution representations, which are notably high for all tokens except "/e", which does not appear in the dataset. These tokens share highly similar semantics and also exhibit very similar next-token distributions, most frequently followed by "*" or ")". This similarity in next-token distribution leads to strong similarities in their embedding vectors. This example vividly illustrates how data distribution shapes semantic structure within the embedding space, particularly in the case of tokens with high semantic affinity.

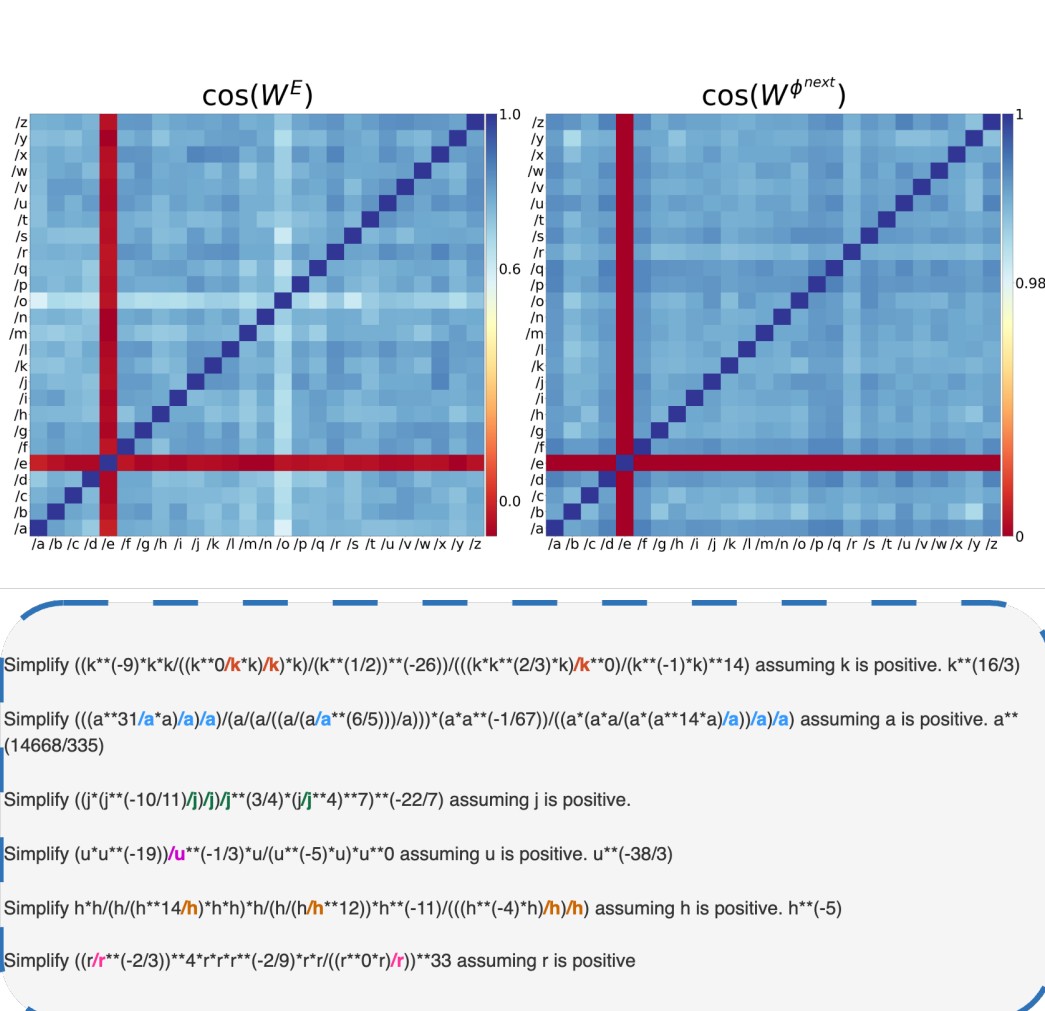

Figure 15: A case analysis of the token group "$/a$", "$/b$", "$/c$", etc. The first row depicts the cosine similarity of their embeddings (left) and distribution representations (right). The second row exhibits the contexts containing these tokens, which are highlighted by different colors.

## C.4 RESULTS OF LLAMA 2

To assess the generalizability of our analysis in Section 6 across different model architectures and tokenizers, we replicate the experiment using the Llama 2 architecture. We employ the same dataset from Pile, and the training configurations are the same as the experiments of Qwen2.5. As shown in Figure 16, the probability signatures effectively capture structural relationships in the embedding space, especially in regions exhibiting high embedding similarity. These results align closely with those in Figure 6, indicating that our analytical approach is robust to variations in model architecture.

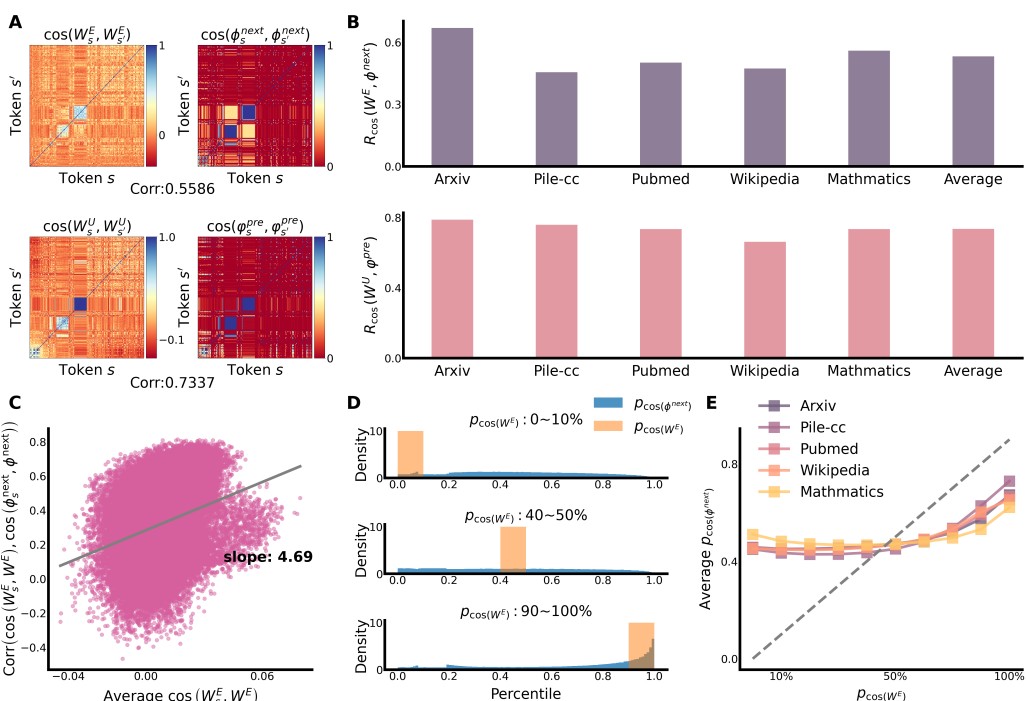

Figure 16: Results with Llama-2 architecture. A: Heatmap of the cosine similarity of $\boldsymbol{W}^E, \boldsymbol{W}^U, \phi^{\text{next}}$ and $\boldsymbol{\varphi}^{\text{pre}}$. B: $R_{\cos}\left(\boldsymbol{W}^E, \phi^{\text{next}}\right)$ (top) and $R_{\cos}\left(\boldsymbol{W}^U, \boldsymbol{\varphi}^{\text{pre}}\right)$ (bottom) with different datasets. C: Relation between $\text{Corr}\left(\cos\left(\boldsymbol{W}_s^E, \boldsymbol{W}^E\right), \cos\left(\phi_s^{\text{next}}, \phi^{\text{next}}\right)\right)$ and the average value of $\cos\left(\boldsymbol{W}_s^E, \boldsymbol{W}^E\right)$. Each point denotes a token $s$. D: Distribution of $p_{\cos(\phi^{\text{next}})}$, conditioned on intervals $0 \sim 10\%, 40 \sim 50\%$ and $90 \sim 100\%$ of the $p_{\cos(\boldsymbol{W}^E)}$. E: Average value of $p_{\cos(\phi^{\text{next}})}$ within each interval of $p_{\cos(\boldsymbol{W}^E)}$.

# D  THEORETICAL DETAILS

## D.1  PROOF OF PROPOSITION 1

**Lemma 1.** *Given a model $F$ and data pair $(\boldsymbol{X}, y) \in \mathbb{N}^{+,L} \times \mathbb{N}^{+}$, $\ell = -\log Softmax\left(F\left(\boldsymbol{X}\right)\right)_y$, we have that*

$$\frac{\partial \ell}{\partial F\left(\boldsymbol{X}\right)} = \boldsymbol{p} - \boldsymbol{e}_y, \tag{5}$$

*where $\boldsymbol{p} = softmax\left(\boldsymbol{X}\right)$.*

*Proof.* It's noted $\ell = -F\left(\boldsymbol{X}\right)_y + \log \sum_{j=1}^{d_{\text{vob}}} \exp F\left(\boldsymbol{X}\right)_j$, then we have

$$\frac{\partial \ell}{\partial F\left(X\right)_i} = -\delta_{i=y} + \frac{\exp F\left(\boldsymbol{X}\right)_i}{\sum_{j=1}^{d_{\text{vob}}} \exp F\left(\boldsymbol{X}\right)_j} = \boldsymbol{p}_i - \delta_{i=y},$$

where $\delta_{i=y} = 1$ if $i = y$ else 0. This indicates that $\frac{\partial \ell}{\partial F(\boldsymbol{X})} = \boldsymbol{p} - \boldsymbol{e}_y$. $\qquad\square$

With Lemma 1, we could obtain the derivative of $\ell$ with respect to $\boldsymbol{W}_x^E$ for any $x \in \mathcal{V}$ as follows:

$$\begin{aligned}
\frac{\partial \ell^i}{\partial \boldsymbol{W}_x^E} &= \frac{\partial F\left(\boldsymbol{X}^i\right)}{\partial \boldsymbol{W}_x^E} \frac{\partial \ell^i}{\partial F\left(\boldsymbol{X}^i\right)} \\
&= \left(\boldsymbol{W}^{U,T}\left(\boldsymbol{p}^i - \boldsymbol{e}_{y^i}\right)\right) \odot G^{(1)}\left(\boldsymbol{W}_{\boldsymbol{X}^i}^E\right).
\end{aligned}$$

Then the gradient flow of $\boldsymbol{W}_x^E$ could be obtained by

$$\frac{d\boldsymbol{W}_x^E}{dt} = -\frac{1}{N} \sum_{i=1}^{N} \frac{\partial \ell^i}{\partial \boldsymbol{W}_x^E} = \frac{1}{N} \sum_{i=1}^{N} \left(\boldsymbol{W}^{U,T}\left(\boldsymbol{p}^i - \boldsymbol{e}_{y^i}\right)\right) \odot G^{(1)}\left(\boldsymbol{W}_{\boldsymbol{X}^i}^E\right),$$

Since $\text{diag}\left(G^{(1)}\left(\boldsymbol{W}_{\boldsymbol{X}^i}^E\right)\right) = 0$ if $x \notin \boldsymbol{X}^i$, we have that

$$\begin{aligned}
\frac{d\boldsymbol{W}_x^E}{dt} &= \frac{1}{N} \sum_{i=1}^{N_x^{\text{in}}} \left(\boldsymbol{W}^{U,T}\left(\boldsymbol{e}_{y_x^i} - \boldsymbol{p}_x^i\right)\right) \odot G^{(1)}\left(\boldsymbol{W}_{\boldsymbol{X}_x^i}^E\right) \\
&= \frac{r_x^{\text{in}}}{N_x^{\text{in}}} \sum_{i=1}^{N_x^{\text{in}}} \left(\boldsymbol{W}^{U,T}\left(\boldsymbol{e}_{y_x^i} - \boldsymbol{p}_x^i\right)\right) \odot G^{(1)}\left(\boldsymbol{W}_{\boldsymbol{X}_x^i}^E\right).
\end{aligned}$$

Since that $y_x^i$ takes value $\nu \in \mathcal{V}$, we can rewrite this formation as

$$\begin{aligned}
\frac{d\boldsymbol{W}_x^E}{dt} &= \frac{r_x^{\text{in}}}{N_x^{\text{in}}} \left[\sum_{\nu \in \mathcal{V}} \sum_{i=1}^{N_{x,\nu}} \left(\boldsymbol{W}^{U,T}\boldsymbol{e}_\nu\right) \odot G^{(1)}\left(\boldsymbol{W}_{\boldsymbol{X}_{x,\nu}^i}^E\right) - \sum_{i=1}^{N_x^{\text{in}}} \left(\boldsymbol{W}^{U,T}\boldsymbol{p}_x^i\right) \odot G^{(1)}\left(\boldsymbol{W}_{\boldsymbol{X}_x^i}^E\right)\right] \\
&= r_x^{\text{in}} \left[\sum_{\nu \in \mathcal{V}} \left(\boldsymbol{W}^{U,T}\boldsymbol{e}_\nu\right) \odot \frac{N_{x,\nu}}{N_x^{\text{in}}} \frac{1}{N_{x,\nu}} \sum_{i=1}^{N_{x,\nu}} G^{(1)}\left(\boldsymbol{W}_{\boldsymbol{X}_{x,\nu}^i}^E\right) - \frac{1}{N_x^{\text{in}}} \sum_{i=1}^{N_x^{\text{in}}} \left(\boldsymbol{W}^{U,T}\boldsymbol{p}_x^i\right) \odot G^{(1)}\left(\boldsymbol{W}_{\boldsymbol{X}_x^i}^E\right)\right],
\end{aligned}$$

where $N_x^{\text{in}}, N_{x,\nu}$ denotes the count of sequences containing $x$ and the count of sequences containing $x$ with label $\nu$, $r_x^{\text{in}} = \frac{N_x^{\text{in}}}{N}, r_{x,\nu} = \frac{N_{s,\nu}}{N}$. Then let $N \to \infty$, by the law of large number we have

$$\begin{aligned}
\frac{d\boldsymbol{W}_x^E}{dt} = r_x^{\text{in}} \Bigg( &\sum_{\nu \in \mathcal{V}} \mathbb{P}_\pi\left(y = \nu \mid x \in \boldsymbol{X}\right)\left(\boldsymbol{W}^{U,T}\boldsymbol{e}_\nu\right) \odot \mathbb{E}_\pi\left[G^{(1)}\left(\boldsymbol{W}_{\boldsymbol{X}}^E\right) \mid x \in \boldsymbol{X}, y = \nu\right] \\
&- \mathbb{E}_\pi\left[\left(\boldsymbol{W}^{U,T}\boldsymbol{p}\right) \odot G^{(1)}\left(\boldsymbol{W}_{\boldsymbol{X}}^E\right) \mid x \in \boldsymbol{X}\right]\Bigg).
\end{aligned}$$

## D.2 Proof of Proposition 2

Similar with the analysis of $\boldsymbol{W}_x^E$, we derive the gradient flow of $\boldsymbol{W}_\nu^U$ as follows:

$$
\frac{d\boldsymbol{W}_\nu^U}{dt} = -\frac{1}{N} \sum_{i=1}^{N} \frac{\partial \ell^i}{\partial \boldsymbol{W}_\nu^U}
$$

$$
= \frac{1}{N} \sum_{i=1}^{N} \left( \boldsymbol{e}_{y^{i},\nu} - \boldsymbol{p}^{i,\nu} \right) \left[ G\left( \boldsymbol{W}_{\boldsymbol{X}^i}^E \right) \right]^T .
$$

Since $\boldsymbol{e}_{y^{i},\nu} = 1$ if $y^i = \nu$ else 0, we have that

$$
\frac{d\boldsymbol{W}_\nu^U}{dt} = \frac{r_\nu^{\text{out}}}{N_\nu^{\text{out}}} \sum_{i=1}^{N_\nu^{\text{out}}} \left[ G\left( \boldsymbol{W}_{\boldsymbol{X}_{(\cdot,\nu)}^i}^E \right) \right]^T - \frac{1}{N} \sum_{i=1}^{N} \boldsymbol{p}^{i,\nu} \left[ G\left( \boldsymbol{W}_{\boldsymbol{X}^i}^E \right) \right]^T ,
$$

where $N_\nu^{\text{out}}$ denotes the count of sequences with label $\nu$ and $r_{\nu_j}^{\text{out}} = \frac{N_\nu^{\text{out}}}{N}$. Then let $N \to \infty$, by the law of large number we have

$$
\frac{d\boldsymbol{W}_\nu^U}{dt} = r_\nu^{\text{out}} \mathbb{E}_\pi \left[ G\left( \boldsymbol{W}_{\boldsymbol{X}}^E \right)^T \mid y = \nu \right] - \mathbb{E}_\pi \left[ \boldsymbol{p}_\nu G\left( \boldsymbol{W}_{\boldsymbol{X}}^E \right)^T \right] .
$$

## D.3 Proof of Corollary 1

With proposition 1, we have that

$$
\frac{d\boldsymbol{W}_x^E}{dt} = r_x^{\text{in}} \left( \sum_{\nu \in \mathcal{V}} \mathbb{P}_\pi \left( y = \nu \mid x \in \boldsymbol{X} \right) \left( \boldsymbol{W}^{U,T} \boldsymbol{e}_\nu \right) \odot \mathbb{E}_\pi \left[ G^{(1)} \left( \boldsymbol{W}_{\boldsymbol{X}}^E \right) \mid x \in \boldsymbol{X} \right] \right.
$$

$$
\left. - \mathbb{E}_\pi \left[ \left( \boldsymbol{W}^{U,T} \boldsymbol{p} \right) \odot G^{(1)} \left( \boldsymbol{W}_{\boldsymbol{X}}^E \right) \mid x \in \boldsymbol{X} \right] \right) .
$$

For the linear model, we have that $G^{(1)} \left( \boldsymbol{W}_{\boldsymbol{X}}^E \right) = \boldsymbol{1}$ if $x \in \boldsymbol{X}$. Utilizing that $\text{softmax} \left( \boldsymbol{f} \right) = \frac{1}{d_{\text{vob}}} \boldsymbol{1} + \frac{1}{d_{\text{vob}}} \boldsymbol{f} + \mathcal{O} \left( d_{\text{vob}}^{-2} \boldsymbol{f} \right)$, we obtain that

$$
\frac{d\boldsymbol{W}_x^E}{dt} = \boldsymbol{W}^{U,T} r_x^{\text{in}} \left( \sum_{\nu \in \mathcal{V}} \mathbb{P}_\pi \left( y = \nu \mid x \in \boldsymbol{X} \right) \boldsymbol{e}_\nu - \mathbb{E}_\pi \left[ \boldsymbol{p} \mid x \in \boldsymbol{X} \right] \right)
$$

$$
= \boldsymbol{W}^{U,T} r_x^{\text{in}} \left( \boldsymbol{\phi}_x^y - \mathbb{E}_\pi \left[ \frac{1}{d_{\text{vob}}} \boldsymbol{1} + \frac{1}{d_{\text{vob}}} \boldsymbol{W}^U \sum_{x_i \in \boldsymbol{X}} \boldsymbol{W}_{x_i}^E + \mathcal{O} \left( d_{\text{vob}}^{-2} \boldsymbol{W}^U \boldsymbol{W}_x^E \right) \mid x \in \boldsymbol{X} \right] \right)
$$

$$
= \boldsymbol{W}^{U,T} r_x^{\text{in}} \left( \boldsymbol{\phi}_x^y - \frac{1}{d_{\text{vob}}} \boldsymbol{1} - \frac{1}{d_{\text{vob}}} \boldsymbol{W}^U \mathbb{E}_\pi \left[ \sum_{x_i \in \boldsymbol{X}} \boldsymbol{W}_{x_i}^E \mid x \in \boldsymbol{X} \right] + \mathcal{O} \left( d_{\text{vob}}^{-2} \boldsymbol{W}^U \boldsymbol{W}_x^E \right) \right)
$$

$$
= \boldsymbol{W}^{U,T} r_x^{\text{in}} \left( \boldsymbol{\phi}_x^y - \frac{1}{d_{\text{vob}}} \boldsymbol{1} - \frac{1}{d_{\text{vob}}} \boldsymbol{W}^U \sum_{x' \in \mathcal{V}} \mathbb{P}_\pi \left( x' \in \boldsymbol{X} \mid x \in \boldsymbol{X} \right) \boldsymbol{W}_{x'}^E + \mathcal{O} \left( d_{\text{vob}}^{-2} \boldsymbol{W}^U \boldsymbol{W}_x^E \right) \right)
$$

$$
= \boldsymbol{W}^{U,T} r_x^{\text{in}} \left( \boldsymbol{\phi}_x^y - \frac{1}{d_{\text{vob}}} \boldsymbol{W}^U \boldsymbol{W}^E \boldsymbol{\phi}_x^{\boldsymbol{X}} - \frac{1}{d_{\text{vob}}} \boldsymbol{1} + \mathcal{O} \left( d_{\text{vob}}^{-2} \boldsymbol{W}^U \boldsymbol{W}_x^E \right) \right)
$$

$$
:= \boldsymbol{W}^{U,T} r_x^{\text{in}} \left( \boldsymbol{\phi}_x^y - \frac{1}{d_{\text{vob}}} \boldsymbol{W}^U \boldsymbol{W}^E \boldsymbol{\phi}_x^{\boldsymbol{X}} + \boldsymbol{\eta} \right) ,
$$

where $\boldsymbol{\eta} = -\frac{1}{d_{\text{vob}}} \boldsymbol{1} + \mathcal{O} \left( d_{\text{vob}}^{-2} \boldsymbol{W}^U \boldsymbol{W}_\alpha^E \right)$ contains the higher-order term and the data independent term.

### D.4 PROOF OF COROLLARY 2

*Proof.* Since the small initialization, we assume that the activation function can be approximated by the following form with the Weierstrass approximation theorem.

$$
\sigma\left(\sum_{x \in \boldsymbol{X}} \boldsymbol{W}_x^E\right) = C_0 + C_1\left(\sum_{x \in \boldsymbol{X}} \boldsymbol{W}_x^E\right) + C_2\left(\sum_{x \in \boldsymbol{X}} \boldsymbol{W}_x^E\right)^{\odot 2} + \boldsymbol{\epsilon}.
$$

With the loss of the generalization, we assume that $C_0 = 0, C_1 = 1, C_2 = \frac{1}{2}$. Then we have

$$
\frac{d\boldsymbol{W}_x^E}{dt} = r_x^{\text{in}} \underbrace{\sum_{\nu \in \mathcal{V}} \mathbb{P}_\pi\left(y = \nu \mid x \in \boldsymbol{X}\right)\left(\boldsymbol{W}^{U,T}\boldsymbol{e}_\nu\right) \odot \mathbb{E}_\pi\left[\boldsymbol{1} + \sum_{x' \in \boldsymbol{X}} \boldsymbol{W}_{x'}^E \mid x \in \boldsymbol{X}, y = \nu\right]}_{\boldsymbol{J}^y}
$$

$$
- r_x^{\text{in}} \underbrace{\mathbb{E}_\pi\left[\left(\boldsymbol{W}^{U,T}\boldsymbol{p}\right) \odot \left(\boldsymbol{1} + \sum_{x' \in \boldsymbol{X}} \boldsymbol{W}_{x'}^E\right) \mid x \in \boldsymbol{X}\right]}_{\boldsymbol{J}^p}.
$$

For the term $\boldsymbol{J}^y$ we have

$$
\boldsymbol{J}^y = \boldsymbol{W}^{U,T} \sum_{\nu \in \mathcal{V}} \mathbb{P}_\pi\left(y = \nu \mid x \in \boldsymbol{X}\right)\boldsymbol{e}_\nu + \sum_{\nu \in \mathcal{V}} \mathbb{P}_\pi\left(y = \nu \mid x \in \boldsymbol{X}\right)\left(\boldsymbol{W}^{U,T}\boldsymbol{e}_\nu\right) \odot \mathbb{E}_\pi\left[\sum_{x' \in \boldsymbol{X}} \boldsymbol{W}_{x'}^E \mid x \in \boldsymbol{X}, y = \nu\right]
$$

$$
= \boldsymbol{W}^{U,T}\boldsymbol{\phi}_x^y + \sum_{\nu \in \mathcal{V}} \text{diag}\left(\boldsymbol{W}_\nu^U\right) \sum_{x' \in \mathcal{V}} \mathbb{P}_\pi\left(y = \nu \mid x \in \boldsymbol{X}\right)\mathbb{P}_\pi\left(x' \in \boldsymbol{X} \mid x \in \boldsymbol{X}, y = \nu\right)\boldsymbol{W}_{x'}^E.
$$

Since that $\mathbb{P}_\pi\left(y = \nu \mid x \in \boldsymbol{X}\right)\mathbb{P}_\pi\left(x' \in \boldsymbol{X} \mid x \in \boldsymbol{X}, y = \nu\right) = \mathbb{P}_\pi\left(x' \in \boldsymbol{X}, y = \nu \mid x \in \boldsymbol{X}\right)$, we have that

$$
\boldsymbol{J}^y = \boldsymbol{W}^{U,T}\boldsymbol{\phi}_x^y + \sum_{\nu, x' \in \mathcal{V}} \mathbb{P}_\pi\left(x' \in \boldsymbol{X}, y = \nu \mid x \in \boldsymbol{X}\right)\boldsymbol{W}_\nu^U \odot \boldsymbol{W}_{x'}^E
$$

$$
= \boldsymbol{W}^{U,T}\boldsymbol{\phi}_x^y + \mathbb{T} \odot \boldsymbol{\phi}_x^{\boldsymbol{X}|y},
$$

where $\mathbb{T} \in \mathbb{R}^{d \times d_{\text{vob}} \times d_{\text{vob}}}$, $\mathbb{T}_{:,x',\nu} = \boldsymbol{W}_\nu^U \odot \boldsymbol{W}_{x'}^E$ for $\nu, x' \in \mathcal{V}$ and $0$ otherwise.

Similarly, for the term $\boldsymbol{J}^p$, we have that

$$
\boldsymbol{J}^p = \mathbb{E}_\pi\left[\left(\boldsymbol{W}^{U,T}\left(\frac{1}{d_{\text{vob}}}\boldsymbol{1} + \frac{1}{d_{\text{vob}}}\boldsymbol{W}^U \sum_{x' \in \boldsymbol{X}} \boldsymbol{W}_{x'}^E\right)\right) \odot \left(\boldsymbol{1} + \sum_{x' \in \boldsymbol{X}} \boldsymbol{W}_{x'}^E\right) \mid x \in \boldsymbol{X}\right]
$$

$$
= \frac{1}{d_{\text{vob}}}\boldsymbol{W}^{U,T}\boldsymbol{1} + \frac{1}{d_{\text{vob}}}\boldsymbol{W}^{U,T} \sum_{x' \in \mathcal{V}} \mathbb{P}_\pi\left(x' \in \boldsymbol{X} \mid x \in \boldsymbol{X}\right)\boldsymbol{W}_{x'}^E + \boldsymbol{\epsilon}
$$

$$
= \frac{1}{d_{\text{vob}}}\boldsymbol{W}^{U,T}\left(\boldsymbol{1} + \boldsymbol{W}^E\boldsymbol{\phi}_x^{\boldsymbol{X}}\right) + \boldsymbol{\epsilon},
$$

where $\boldsymbol{\epsilon} = \mathcal{O}\left(\frac{1}{d_{\text{vob}}^2}\boldsymbol{W}^U\boldsymbol{W}_\alpha^E\right)$. Then we have that

$$
\frac{d\boldsymbol{W}_\alpha^E}{dt} = r_x^{\text{in}}\left(\boldsymbol{W}^{U,T}\boldsymbol{\phi}_x^y - \frac{1}{d_{\text{vob}}}\boldsymbol{W}^{U,T}\boldsymbol{W}^E\boldsymbol{\phi}_x^{\boldsymbol{X}} + \mathbb{T} \cdot \boldsymbol{\phi}_x^{\boldsymbol{X}|y} + \boldsymbol{\epsilon}\right),
$$

where $\boldsymbol{\epsilon} = -\frac{1}{d_{\text{vob}}}\boldsymbol{W}^{U,T}\boldsymbol{1} + \mathcal{O}\left(\frac{1}{d_{\text{vob}}^2}\boldsymbol{W}^U\boldsymbol{W}_\alpha^E\right)$. $\qquad\square$

## D.5 PROOF OF COROLLARY 3

*Proof.* With Proposition 2, we have that

$$\frac{d\boldsymbol{W}_\nu^U}{dt} = \frac{r_\nu^{\text{out}}}{N_\nu^{\text{out}}} r_\nu^{\text{out}} \mathbb{E}_\pi \left[ \left( \sum_{x \in \boldsymbol{X}} \boldsymbol{W}_x^E \right)^T \mid y = \nu \right] - \mathbb{E}_\pi \left[ \boldsymbol{p}_\nu \left( \sum_{x \in \boldsymbol{X}} \boldsymbol{W}_x^E \right)^T \right]$$

$$= L r_\nu^{\text{out}} \sum_{x \in \mathcal{V}} \mathbb{P}_\pi \left( x \in \boldsymbol{X} \mid y = \nu \right) \boldsymbol{W}_x^{E,T} - \frac{1}{d_{\text{vob}}} \boldsymbol{W}^E \mathbf{1} + \boldsymbol{\epsilon}$$

$$= L r_\nu^{\text{out}} \left( \boldsymbol{W}^E \boldsymbol{\varphi}_\nu^{\boldsymbol{X}} \right)^T - \boldsymbol{\eta},$$

where $\boldsymbol{\eta} = -\frac{1}{d_{\text{vob}}} \boldsymbol{W}^E \mathbf{1} + \mathcal{O} \left( \frac{1}{d_{\text{vob}}} \boldsymbol{W}^E \boldsymbol{W}^E \mathbf{1} \right)$. □

## D.6 PROOF OF COROLLARY 4

*Proof.* The next-token-prediction training loss could be formulated as

$$\ell^i = \frac{1}{L} \sum_{t=1}^{L-1} \text{CrossEntropy} \left( F_{\text{lan}} \left( \boldsymbol{X}_{:t} \right); \boldsymbol{e}_{\boldsymbol{X}_{t+1}} \right).$$

So we have that

$$\frac{\partial \ell^i}{\partial \boldsymbol{W}_s^E} = \frac{1}{L} \sum_{t=1}^{L-1} \boldsymbol{W}^{U,T} \left( \boldsymbol{p}_t^i - \boldsymbol{e}_{\boldsymbol{X}_{t+1}^i} \right) \odot \left( \delta_{\boldsymbol{X}_t^i = s} \mathbf{1} + \tilde{F}^{(1)} \left( \boldsymbol{X}_{:t}^i \right) \right).$$

Furthermore, we have that

$$\frac{d\boldsymbol{W}_s^E}{dt} = \frac{1}{NL} \sum_{i=1}^N \sum_{t=1}^{L-1} \boldsymbol{W}^{U,T} \left( \boldsymbol{e}_{\boldsymbol{X}_{t+1}^i} - \boldsymbol{p}_t^i \right) \odot \left( \delta_{\boldsymbol{X}_t^i = s} \mathbf{1} + \tilde{F}^{(1)} \left( \boldsymbol{X}_{:t}^i \right) \right)$$

$$= \frac{1}{NL} \boldsymbol{W}^{U,T} \sum_{i=1}^N \sum_{t=1}^{L-1} \delta_{\boldsymbol{X}_t^i = s} \boldsymbol{e}_{\boldsymbol{X}_{t+1}^i} + \frac{1}{NL} \boldsymbol{W}^{U,T} \sum_{i=1}^N \sum_{t=1}^{L-1} \boldsymbol{e}_{\boldsymbol{X}_{t+1}^i} \odot \tilde{F}^{(1)} \left( \boldsymbol{X}_{:t}^i \right)$$

$$- \frac{1}{NL} \sum_{i=1}^N \sum_{t=1}^{L-1} \boldsymbol{W}^{U,T} \boldsymbol{p}_t^i \odot \left( \delta_{\boldsymbol{X}_t^i = s} \mathbf{1} + \tilde{F}^{(1)} \left( \boldsymbol{X}_{:t}^i \right) \right).$$

Since the small initialization, assuming that $||\boldsymbol{W}||_\infty = \mathcal{O} \left( d^{-\gamma} \right)$ for any trainable parameter matrix $\boldsymbol{W}$, we have that $||\tilde{F}^{(1)} \left( \boldsymbol{X}_{:t}^i \right)||_\infty = \mathcal{O} \left( d^{1-2\gamma} \right)$ in the initial stage. Let $N \to \infty$, we have that

$$\frac{d\boldsymbol{W}_s^E}{dt} = r_s^{\text{in}} \boldsymbol{W}^{U,T} \left( \boldsymbol{\phi}_s^{\text{next}} - \boldsymbol{\eta}^E \right),$$

where $\boldsymbol{\eta}^E = \sum_{t=1}^{L-1} \mathbb{E}_\pi \left[ \boldsymbol{p} \mid \boldsymbol{X}_t = s \right] + \mathcal{O} \left( d^{1-2\gamma} \boldsymbol{\phi}_s^{\text{next}} \right)$. Similarly, we have that

$$\frac{d\boldsymbol{W}_s^U}{dt} = \frac{1}{NL} \sum_{i=1}^N \sum_{t=1}^{L-1} \left( \delta_{\boldsymbol{X}_{t+1}^i = s} - \boldsymbol{p}_{\boldsymbol{X}_{:t}^i}^{i,s} \right) \left( \boldsymbol{W}_{\boldsymbol{X}_t^i}^{E,T} + \tilde{F} \left( \boldsymbol{X}_{:t}^i \right)^T \right),$$

where $\boldsymbol{p}_{\boldsymbol{X}_{:t}^i}^{i,s}$ means the $s$-th element of the output probability with input sequence $\boldsymbol{X}_{:t}^i$. Let $N \to \infty$, we have

$$\frac{d\boldsymbol{W}_s^U}{dt} = r_s^{\text{out}} \left( \boldsymbol{W}^E \boldsymbol{\varphi}_s^{\text{pre}} \right)^T + \boldsymbol{\eta}^U,$$

where $\boldsymbol{\eta}^U = \sum_{t=1}^{L-1} \mathbb{E}_\pi \left[ \boldsymbol{p}_{\boldsymbol{X}_{:t}}^s \boldsymbol{W}_{\boldsymbol{X}_t}^{E,T} \right] + \mathcal{O} \left( r_s^{\text{out}} d^{1-2\gamma} \left( \boldsymbol{W}^E \boldsymbol{\varphi}_s^{\text{pre}} \right)^T \right)$. □

