# OpenReview forum: "Probability Signature: Bridging Data Semantics and Embedding Structure in Language Models"
_ICLR.cc/2026/Conference — Submitted to ICLR 2026_

### Official Review · Reviewer_khiJ · 2025-10-28

**Soundness:** 2
**Presentation:** 2
**Contribution:** 2
**Rating:** 4
**Confidence:** 4

**Summary:**

This paper considers embedding models such as MLPs and LLMs, and to shred light on how the structure of the embedding space relates to the model and the data generating distribution.

They do this by defining probability signatures which represent properties of the data generating distribution (though this is not entirely clear, see Weaknesses) and showing how the gradient flow for the embedding and unembedding matrices depends on them for a collection of models (linear, MLP, LLM).

Unfortunately, it is not clear precisely what mathematical objects the probability signatures are, making it hard to understand the implications of the results (see Weaknesses)

Experimentally, the authors plot matrices of distances between columns of the embedding  / unembedding matrix and of ground-truth distances between the tokens in their experiments. They recorded how correlations between these matrices change over training, and they are typically highly correlated after training when the model is sufficiently expressive to represent the true label function.

**Strengths:**

I think this paper is potentially full of good ideas, but the lack of accuracy in many places makes it hard to know.

With improved accuracy, I think Propositions 1 and 2 have real potential value. They show the form of the gradient flow of the embedding and unembedding matrices, and the proofs appear to be sound (although pointing the proofs in the main text would be nice).

I also think that the corollaries which show simplified forms of these flows for specific models could be very valuable.

**Weaknesses:**

I have two main strands of comments, one of the implications of the results, and the other on accuracy.

**Implications.**
The central message of this paper is not clear. The paper would really benefit from clearly stating its goal, conclusions, and how the theoretical results and experiments back up the conclusion. I think there's potential good ideas in this paper, but its hard to see how they relate to one another.

While I think the derivation of the gradient flows could be a valuable contribution by itself, its not clear to me from the paper what this actually tells us about the geometry of the embedding space, and how they relate to the experiments and plots in the paper. Some additional explanation would be helpful.

**Accuracy.**
There are many parts of the paper where a lack of mathematical accuracy makes it hard to know what is being said.

- The definitions of $\phi_x^y, \phi_x^X$ and $\psi_\nu^X$ define vectors and the definition of $\phi_x^{X \mid y}$ defines a matrix. Yet, in the text immediately below $\phi_x^y$ is described as a distribution and $\phi_x^X$ as a probability (i.e. a scalar). Figure 1 shows plots of $\phi_\alpha^X$ which without further explanation suggests they are functions (perhaps these are supposed to be pdfs of distributions?).
- In Proposition 1, $N_x^{\text{in}}$ and $N_{x,\nu}$ are defined as "the count of sequences contain $x$....". I don't know from context what  I am counting in. Is it the batch or something else?
- In Section 5, function $f_{\text{add}}. \tilde{f}_{\text{add}},\ldots$ etc. are defined. I assume these define the labels $y$, but its not said.
- Given I am unclear about what the probability signatures are, its not clear to me what is plotted in Figure 1.
- What is $\cos(x,y)$? Is it the cosine distance between $x$ and $y$? If so, this is not the same as the cosine.

**Questions:**

Please clarify
1) The goal of the paper. Is it to understand the geometry of the embedding space, or is it to understand the training dynamics. If the goal is the latter, what can be learned from the theoretical results and experiments?
2) The definitions of the quantities I mention in the weaknesses section.
3) What is being plotted in Figure 1?

---

> ### Author Response · Authors · 2025-11-19
> **Reply to Reviewer khiJ (1/2)**
>
> We sincerely thank you for reviewing our manuscript. We have carefully considered all your feedback and thoroughly revised the paper accordingly. Below, we provide detailed responses to each specific comment, along with the corresponding modifications:
> ***
> ***Weakness about Implications***: We sincerely thank the reviewer for identifying the weaknesses in our writing. We have completely reorganized and rewritten the structure of the main text, abandoning our previous approach. Instead, we now present our work step-by-step according to our research process, clearly explaining each component and its motivation.
>
> Please allow me to briefly introduce our research motivation, content, and contributions:
>
> >Current development of LLMs overly relies on increasing model parameters and data volume, making the development costs increasingly prohibitive, which is an unscientific and unsustainable path (Introduction, Paragraph 1). To break this dilemma, a key solution lies in understanding how current models actually learn and operate. This is also the shared goal of numerous interpretability works. Among these, many studies have focused on how the embedding space encodes natural language, finding that the embedding space can learn semantic associations between tokens—for instance, digit embeddings are often arranged following numerical order (Figure 1). Our research objective is to provide an analytical framework that theoretically clarifies why embeddings can learn these associations, with corresponding experimental verification.
> >
> >Through analyzing the gradient flow of embeddings, we discovered how token relationships affect embedding dynamics (Proposition 1). Simultaneously, we found that which token relationships embeddings capture is concretely linked to model architecture. We separately analyzed linear models (Corollary 1) and feedforward networks (Corollary 2), and verified our analysis through controlled experiments. Furthermore, we extend our analysis framework to Transformer-based LM (Section 6). Even with the leading term of the model, the probability signature we uncover could largely dominate the dynamics of the embedding structure. Our key contribution lies in proposing a viable analytical method for embedding structures. Researchers can follow our approach to conduct similar studies on models of their interest, to analyze which token relationships influence the embedding space under that specific model architecture.
> ***
> ***Gradient Flow***: Gradient flow reveals which token associations guide the dynamics of embeddings. For instance, Corollary 1 shows that the gradient flow of a linear model is influenced by $\phi_x^y$ and $\phi_x^X$. Therefore, if in a task two tokens $x$ and $x'$ satisfy $\phi_x^y=\phi^y_{x^\prime}$ and $\phi_x^X=\phi_{x^\prime}^X$, their embedding vectors should also be nearly identical. To this end, we designed controlled variable experiments (Section 4.1 Experimental Validation: Controllable Addition Tasks) to verify our conclusions. The experimental results (Figure 2) demonstrate that if two tokens $x$ and $x'$ in a task satisfy $\phi_x^y=\phi_{x^\prime}^y$ and $\phi_x^X=\phi_{x^\prime}^X$, their embedding vectors indeed converge to the same direction. Additionally, the PCA projection results (Figure 4) further illustrate that the structure of the signature can predict the structure of the embedding space. In the revised manuscript, we have provided clearer explanations and interpretations of the significance of each experiment and result.  At line 245-249, we clarify the motivation of our tasks as follows:
>
> >If two tokens $\alpha,\alpha'$ satisfy $\phi_{\alpha}^y\approx\phi_{\alpha'}^y$ and $\phi_{\alpha}^{X}\approx\phi_{\alpha'}^{X}$, Corollary 1 forces their embeddings to align: $\cos(W^E_{\alpha},W^E_{\alpha'}) \to 1$. We design three variable-controlled addition tasks to isolate and verify each probability signature's influence. In each task, $\phi_\alpha^{y}$ or $\phi_\alpha^{X}$ or both of them will be identical across $\alpha$.
> ***
> ***Accuracy 1 & 4***: We sincerely apologize for the confusion caused by our previous description. In the revised version, we have carefully defined the signature: each signature is a vector or matrix whose values represent a corresponding conditional probability. For example:
>
> - The $i$-th element of $\phi_x^y$ has the value $P(y=i \mid x \in X)$
> - The $i$-th element of $\phi_x^X$ has the value $P(i \in X \mid x \in X)$
> - The value at the $i$-th row and $j$-th column of $\phi_x^{X|y}$ is $P(y=i, j \in X \mid x \in X)$
>
> This is also why we describe the probability signature as a "distribution"—it is essentially a probability feature vector/matrix of a discrete random variable. Figure 1 (now Figure 8) illustrates what these vectors/matrices look like in addition tasks when α takes different values.

---

> > ### Author Response · Authors · 2025-11-19
> > **Reply to Reviewer khiJ (2/2)**
> >
> > ***Accuracy 2***: $N_x^{in}$ denotes the number of sequences in a training set $\{(X^i, y^i)\}$ where $X^i$ is a sequence that contains token $x$. For example, consider the following five sequences: $X^1=[x,y,z,a,b]$, $X^2=[y,y,z,a,b]$, $X^3=[z,y,z,x,b]$, $X^4=[t,y,z,a,b]$, $X^5=[t,y,z,a,x]$. Token $x$ appears only in $X^1$, $X^3$, and $X^5$, so $N_x^{in}=3$.
> >
> >  $N_{x,\nu}$ denotes the number of sequences where $X^i$ contains $x$ and $y^i$ equals $\nu$. For instance, if we assign labels $y^1=\nu$, $y^2=w$, $y^3=w$, $y^4=\nu$, $y^5=\nu$ to the five sequences above, only $X^1$ and $X^5$ satisfy both conditions (containing $x$ and having label $\nu$), so $N_{x,\nu}=2$.
> >
> > We agree that such notation could easily cause confusion. Therefore, we rewrote Proposition 1 and Proposition 2 into their expectation forms, significantly reducing the amount of mathematical notation in the paper. We have provided definitions and explanations for every symbol; in Section 4, we include a dedicated explanation of gradient flow before its introduction to alleviate reader difficulty.
> > ***
> > ***Accuracy 3***: Thanks for your feedback. Yes, for each task we define $y=f(X)$, and we have made this explicit in the updated manuscript.
> > ***
> > ***Accuracy 5***: We realized that our previous notation was not clear enough, so we removed those symbols and used the most primitive and simplest notation for representation.
> >
> > Taking Figure 2 as an example, the $\cos(\phi_\alpha^y, \phi_\alpha^y)$ in the leftmost first column represents the cosine similarity between the two vectors $\phi_\alpha^y$ and $\phi_\alpha^y$. Each figure row represents a different task. Under our experimental setup, there are 10 different $\alpha$ values, so each figure displays a $10 \times 10$ matrix. The element at row $i$ and column $j$ of this matrix represents the cosine similarity between probability signatures of the $i$-th $\alpha'$ and the $j$-th $\alpha$, with its value indicated by color (the bluer it is, the closer it is to 1).
> > ***
> > We sincerely thank you once again for your invaluable comments. We hope that our revisions and responses could address your concerns. We wish you the very best.

---

> > > ### Comment · Reviewer_khiJ · 2025-11-24
> > >
> > > Thank you for the clarifications. The writing in this revision does seem greatly improved, and I'm glad that the review process has improved the paper. Given the nature of issues at the initial stage of the review process, I think this paper requires a full re-review. I think there are good ideas here, and I would encourage the authors to carefully consider the comments of the reviewers, ensure that all ideas are communicated clearly and accurately, and resubmit their paper. For this reason, I recommend rejection at this stage.

---

### Official Review · Reviewer_PGHA · 2025-11-01

**Soundness:** 2
**Presentation:** 1
**Contribution:** 1
**Rating:** 2
**Confidence:** 3

**Summary:**

This paper studies the relationships between probabilistic structures in a model's training data, the gradients used to train it, and the resulting model weights. By defining probability signatures the authors study when ordinal structure emerges in the model trained on an addition task. The analysis is then further extended to a Language model trained on the pile dataset, showing how embedding structure changes over training.

**Strengths:**

The paper looks at a worthwhile question - understanding the relation between data structure, gradient structure and weight structure. While extensive interpretability work has look at model weights, far less relates this to gradient information in a meaningful way.

**Weaknesses:**

This paper is overall challenging to read. In part because it would benefit from a proofread for grammatical and typographic errors. In part because the authors choices of formalisation, which are verbose and at times difficult to follow. Additionally visuals often lack adequate labelling - figure 1 for example does not indicate the units for the x axis.

More generally the novelty of the work is currently unclear. The majority of results appear to show relationships between structure in the training data and in the embedding matrix, an area that is relatively well studied. Additionally the core claim appears to be that the statistical structure of the training data affects the solution a model learns; this is a foundational premise of machine learning. Core results appear to be presented in figures 4 and 6 as heatmaps with no summary statistics - it is difficult to tell the significance of a result from looking at a heatmap alone.

If the authors could better clarify how their work studies gradient information, which they claim it does, that may help support novelty here.

**Questions:**

Is this work's core finding that the statistical structure of the data affects learning?

You mention semantics in the title of the paper and the introduction. However the work itself makes no mention of the discipline of semantics in NLP, Neuroscience, Linguistics, or Cognitive Science more generally. What is the definition of semantics here and how does it relate to other fields?

---

> ### Author Response · Authors · 2025-11-19
> **Reply to Reviewer PGHA (1/2)**
>
> We sincerely thank you for taking the time to read and review our work. We have carefully considered all of your concerns and questions, and have thoroughly revised the manuscript accordingly. We have submitted the revised version and highlighted the core contexts. Below are our detailed responses to each specific comment, along with our modifications:
> ***
> ***Weakness 1***: We sincerely thank the reviewer for identifying the weaknesses in our writing. We have completely reorganized and rewritten the structure of the main text, abandoning our previous approach. Instead, we now present our work step-by-step according to our research process, clearly explaining each component and its motivation. We have moved all supplementary experimental analyses to the appendix, retaining only key results and discussions in the main text, which has substantially improved readability. Additionally, we revised Proposition 1 and Proposition 2 into their expectation forms, significantly reducing mathematical notation. We have also provided clear definitions and explanations for all symbols; in Section 4, we include a dedicated explanation of gradient flow before its introduction to minimize reader confusion.
> ***
> ***Weakness 2 & Question 1***: We sincerely apologize for any confusion or misunderstanding that may have arisen from our previous manuscript. Please allow me to briefly introduce our research motivation, content and contributions:
>
> >Current development of LLMs overly relies on increasing model parameters and data volume, making the development costs increasingly prohibitive, which is an unscientific and unsustainable path (Introduction, Paragraph 1). To break this dilemma, a key solution lies in understanding how current models actually learn and operate. This is also the shared goal of numerous interpretability works. Among these, many studies have focused on how the embedding space encodes natural language, finding that the embedding space can learn semantic associations between tokens—for instance, digit embeddings are often arranged following numerical order (Figure 1). Our research objective is to provide an analytical framework that theoretically clarifies why embeddings can learn these associations, with corresponding experimental verification.
> >
> >Through analyzing the gradient flow of embeddings, we discovered how token relationships affect embedding dynamics (Proposition 1). Simultaneously, we found that which token relationships embeddings capture is concretely linked to model architecture. We separately analyzed linear models (Corollary 1) and feedforward networks (Corollary 2), and verified our analysis through controlled experiments. Furthermore, we extend our analysis framework to Transformer-based LM (Section 6). Even with the leading term of the model, the probability signature we uncover could largely dominate the dynamics of the embedding structure. Our key contribution lies in proposing a viable analytical method for embedding structures. Researchers can follow our approach to conduct similar studies on models of their interest, to analyze which token relationships influence the embedding space under that specific model architecture.
>
> In the revised manuscript, we have emphasized our contributions in the Introduction, Methodology, Results, and Conclusion sections to facilitate reader comprehension.

---

> > ### Author Response · Authors · 2025-11-19
> > **Reply to Reviewer PGHA (2/2)**
> >
> > ***Question 2***: Thank you very much for pointing this out and raising this critical question. In the revised manuscript, we provide a detailed discussion in Section 3.2 on the connection between the token relationships we investigate, our defined probability signature, and semantics:
> >
> > >In natural language, a token's meaning is fully constituted by its statistical context: how it predicts downstream labels, what tokens it co-occurs with, and how these relationships jointly evolve. Formally, these semantic regularities manifest as conditional probability distributions over the vocabulary. For a token $x$ in input $X$, we consider four representative families of such distributions:
> > >
> > >- **Label relationships**: $P_{\pi}(y=\nu\mid x\in X)$ encodes what $x$ signals about the output—e.g., "excellent" in a review robustly predicts a positive label $\nu$, while "frustrated'' skews toward negative.
> > >
> > >- **Co-occurrence relationships**: $P_{\pi}(x^\prime\in X\mid x\in X)$ captures syntactic-semantic neighborhoods—"stock'' frequently co-occurs with "market'' but rarely with "apple'' (in the financial sense). Higher-order terms like $P_{\pi}(x^\prime,x^{\prime\prime}\in X\mid x \in X)$ encode compositional contexts.
> > >- **Joint relationships**: The joint $P_{\pi}(x^\prime \in X, y=\nu\mid x\in X)$ reveals context-dependent labeling—"apple'' co-occurring with "pie'' predicts a food label, while with "store'' predicts a tech label.
> > >- **Inverse relationships**: $P_{\pi}(x^\prime\in X\mid y=x)$ describes what precedes a token as its cause—the tokens that predict $x$ itself (e.g., what contexts make "surprised'' likely to appear).
> > ***
> > We sincerely thank you again for your invaluable comments. We hope our revisions and responses have satisfactorily addressed your concerns, and we wish you all the best.

---

### Official Review · Reviewer_qPqX · 2025-11-01

**Soundness:** 3
**Presentation:** 2
**Contribution:** 3
**Rating:** 4
**Confidence:** 3

**Summary:**

- This paper studies how data distributions shape embedding structures in language models, proposing a collection of probability signatures that encode semantic relationships among token embeddings.
- The authors approach this problem through theoretical gradient flow analysis, small-scale experiments (single-layer linear models and feedforward networks) on toy composite addition tasks, and large-scale experiments using the Qwen2.5 architecture trained on subsets of the Pile dataset (as well as analyzing the pretrained mode)
- They find that embedding similarities align with these probability signatures, revealing how semantic structures emerge from data.

**Strengths:**

- The paper tackles an interesting and fundamental problem of the impact of data distribution on embedding structures and introduces a clear framework for understanding this via probability signatures, providing an interpretable lens into how data semantics are encoded in embeddings.
- It also has strong theoretical grounding, employing gradient flow dynamics to give a principled explanation for the relationship between embeddings and the probability signatures derived from training data.

**Weaknesses:**

- Due to the extensive notation and multiple quantities being analyzed, the paper could benefit from improved clarity and organization, clearly summarizing the main findings and their supporting evidence in each section.
- While the controlled synthetic experiments are valuable, it would strengthen the paper to include additional large-scale experiments in the LLM setting to complement the ones already in the paper.
- For LLMs, since the model consists of multiple transformer blocks that further process the initial embeddings, it would be helpful to extend the analysis beyond the initial embedding layer, as this could provide insight into how these relationships evolve deeper in the network (at least empirically).
- The paper lacks discussion on practical implications; elaborating on how these findings might inform future model design, training strategies, or interpretability efforts would make the work more impactful.

**Questions:**

- Just to clarify, when computing the probability signatures, it is just based on proportion of the occurrences in the train data across all sequences?
- It may be good to check how some of the metrics progress across training (eg. the correlations $R_{\text{cos}}(W^E,\phi^{\text{next}})$ and $R_{\text{cos}}(W^U,\varphi^{\text{pre}})$ as part of your analysis

---

> ### Author Response · Authors · 2025-11-19
> **Reply to Reviewer qPqX (1/2)**
>
> We sincerely thank you for taking the time to review our manuscript. We have carefully considered all your feedback and thoroughly revised the paper accordingly. Below, we provide detailed responses to each specific comment, along with the corresponding modifications:
> ***
> ***Weakness 1***:We sincerely thank the reviewers for identifying the writing deficiencies in our manuscript. In response, we have thoroughly reorganized and rewritten the entire structure of the main text. We abandoned our previous writing approach in favor of presenting each component of our work step-by-step, following our actual research process and explicitly explaining the motivation behind each section.
>
> We have moved all supplementary experimental analyses to the appendix, retaining only key results and essential discussions in the main text. This significantly improves readability for readers. Additionally, we rewrote Propositions 1 and 2 in expectation form, which substantially reduces mathematical notation complexity. Every symbol is now clearly defined upon first use, and we added a dedicated introduction to gradient flow concepts in Section 4 before presenting our formal analysis. These changes collectively make the paper much more accessible while preserving its technical rigor.
>
> Please allow me to briefly introduce our research motivation, content, and contributions:
>
> >Current development of LLMs overly relies on increasing model parameters and data volume, making the development costs increasingly prohibitive, which is an unscientific and unsustainable path (Introduction, Paragraph 1). To break this dilemma, a key solution lies in understanding how current models actually learn and operate. This is also the shared goal of numerous interpretability works. Among these, many studies have focused on how the embedding space encodes natural language, finding that the embedding space can learn semantic associations between tokens—for instance, digit embeddings are often arranged following numerical order (Figure 1). Our research objective is to provide an analytical framework that theoretically clarifies why embeddings can learn these associations, with corresponding experimental verification.
> >
> >Through analyzing the gradient flow of embeddings, we discovered how token relationships affect embedding dynamics (Proposition 1). Simultaneously, we found that which token relationships embeddings capture is concretely linked to model architecture. We separately analyzed linear models (Corollary 1) and feedforward networks (Corollary 2), and verified our analysis through controlled experiments. Furthermore, we extend our analysis framework to Transformer-based LM (Section 6). Even with the leading term of the model, the probability signature we uncover could largely dominate the dynamics of the embedding structure. Our key contribution lies in proposing a viable analytical method for embedding structures. Researchers can follow our approach to conduct similar studies on models of their interest, to analyze which token relationships influence the embedding space under that specific model architecture.
> ***
> ***Weakness 2***: Thank you very much for your guidance and comments on our experimental design. Our primary contribution lies in proposing this research methodology—analyzing embedding structures through gradient flow equations. Through analysis of simplified models and experimental validation, we demonstrate that this research framework is both feasible and accurate. For real language models, by analyzing only the dominant term in the residual stream, the resulting probability signature can already capture a substantial portion of the embedding structure, including the ordered arrangement of digits shown in Figure 1. This precisely proves that our analytical approach is sufficiently effective and reasonable.
> In the language model section, we utilized both Llama-2 and Qwen2.5 architectures with five distinct training datasets each. All ten experimental configurations validated our analysis. We believe these results demonstrate that our analytical framework is applicable across different architectures and datasets.

---

> > ### Author Response · Authors · 2025-11-19
> > **Reply to Reviewer qPqX (2/2)**
> >
> > ***Weakness 3***: We fully agree with your perspective, which will be a key focus of our future work. The main contribution of this paper lies in proposing an approach to analyze how the embedding space captures token associations through gradient flow. This methodology can be extended to any model architecture, allowing researchers to follow our framework to analyze models of their own interest. To illustrate our motivation for analyzing the leading term, we made a clarification in Section 6 as follows:
> >
> > >A full analysis of all terms in Proposition 1 for Transformers would be intractable and, more importantly, unnecessary for validating our core contribution. We therefore adopt a minimalist validation strategy: analyze the dominant probability signature predicted by gradient flow and test whether it alone can predict embedding structure. If this simplified analysis succeeds, it proves that our framework captures the essential mechanism and researchers can then extend it to additional modules as needed.
> > ***
> > ***Weakness 4***: Thank you very much for your valuable feedback. In the revised version, we discuss the implications of our work for model architecture design and loss function design in the second paragraph of Section 7 as follows:
> >
> > >We illustrate that each architecture implicitly selects which probability signatures it can encode. Our gradient-flow analysis makes this selection explicit and quantifiable: Corollary 1 proves that linear models cannot encode joint token-label relationships ($ \phi_x^{X\mid y} $). Any task requiring this relationship will fail, regardless of scale. Adding a nonlinear activation unlocks $\phi_x^{X\mid y}$ (Corollary 2), enabling models to learn such semantics. This suggests a principled architecture search: introduce modules whose Jacobians $G^{(1)}$ encode desired probability signatures. On the other hand, our results have shown that the loss function is not merely a performance metric but also a gradient flow sculptor that determines which probability signatures dominate. Corollary 4 shows that next-token prediction makes $\phi_s^{\rm next}$ the dominant signature, embedding tokens based on immediate neighbors. This explains why standard autoregressive models excel at local coherence but struggle with long-range dependencies. If the loss predicts $k$ future tokens, gradient flow will encode the k-gram relationship distribution. This provides a theoretical explanation for why multi-token prediction could easily capture the global relationships (Gloeckle et al., 2024).
> > ***
> > ***Question 1***: Yes, the calculation of probability features depends only on the training data. For example, for a training dataset $ \\{\left(X^i,y^i\right)\\}$, we calculate $P_\pi(y = \nu | x \in X)$ by counting the proportion of sequences in the training set where $x$ occurs that have label $\nu$ .
> > ***
> > ***Question 2***:Thank you very much for your feedback. We agree that the dynamics of these metrics are essential. We continue the training to 20 epochs in each experiment in Section 6 and provide a dynamics evaluation in revised version.  In Figure 6, we show the evolution of the correlation between the embedding structure and the probability signature structure the whole training, and provide the following analysis in the revised manuscript:
> >
> > >Figure 6 B tracks the $R_{\cos}\left(W^E,\phi^{\rm next}\right)$ and $R_{\rm cos}\left(W^U,\varphi^{\rm pre}\right)$ across all subsets during training. Correlations increase during the first epoch, indicating that gradient flow rapidly encodes next-token and previous-token statistics into embeddings and unembeddings. After reaching peak alignment, correlations plateau and dip slightly, showing that the embedding structure is still largely impacted by $\phi^{\rm next}_s$ and $\varphi^{\rm pre}_s$. The fact that a single simplified probability signature maintains predictive power throughout training, proves that our gradient flow analysis captures the essential mechanism of embedding structure.
> > ***
> > We sincerely thank you once again for your invaluable comments. We hope that our revisions and responses could address your concerns. We wish you the very best.

---

### Official Review · Reviewer_sy2v · 2025-11-01

**Soundness:** 2
**Presentation:** 1
**Contribution:** 2
**Rating:** 4
**Confidence:** 4

**Summary:**

This paper explores how the structure of token embeddings in language models arises from the statistical properties of the training data. The authors propose the concept of  “probability signatures” —statistical descriptors derived from input–output co-occurrence distributions—that aim to explain and predict how embeddings are organized. Using controlled addition tasks with linear and feedforward models, as well as experiments on Qwen2.5 models trained on subsets of *The Pile*, the paper shows strong correlations between embedding similarity and these probability signatures. Theoretical analysis based on gradient flow dynamics further connects data distribution, optimization behavior, and model architecture, offering a unified view of how semantic patterns in data shape embedding geometry.

**Strengths:**

1. The paper introduces the concent of "probability signatures" to links data statistics and the geometric structure of the embedding space, which is interesting.

**Weaknesses:**

1. The writting of this paper should be improved signficantly. The current manuscript does not look like a professional acamedic paper, in terms of organization of the content and clearness of the content. For instances, almost all of the content in the first and many content in second paragraphs are too broad and have no close connection to the topic of the paper, making them look very superfluous. After reading the introduction, it is still not clear why do we need to study this problem and why this problem is important.

2. The poor writting is also reflected in the clearness and rigorousness of the content. Let's take the gradient flow defined in Proposition 1 as an example, the authors never formally introduce the time variable, but they directly define the gradient $\frac{d {\mathbf{W}}_x^E}{d t}$. The presentation here is very confusing and unrigorous.

3. While the paper analyzes how data distribution shapes embeddings, its practical implications for improving or understanding large-scale models are still limited.

4. The theoretical analysis is based on highly simplified models and may not capture the complexity of modern Transformer architectures, so its explanatory power for real LMs’ embeddings is limited.

5. The framework mainly holds under small initialization and early-stage linearized training. It remains unclear whether such simple probability signatures still explain embeddings after nonlinear dynamics dominate.

**Questions:**

1. During LLM training, does the influence of probability signatures on embedding structure emerge rapidly and then stabilize, or does it continue to increase throughout training? Providing an empirical trend or metric over training steps would be helpful.

2. As training proceeds beyond the small-initialization and early-linearization regime, does the explanatory power of probability signatures remain consistent, or does it degrade as nonlinear effects accumulate?

3. Could the proposed probability-signature framework have potential applications in guiding the training of modern LLMs? If so, what are the possible directions or limitations?

---

> ### Author Response · Authors · 2025-11-19
> **Reply to Reviewer sy2v (1/2)**
>
> We sincerely thank you for taking the time to read and review our work. We have carefully considered all of your concerns and questions, and have thoroughly revised the manuscript accordingly. Below are our detailed responses to each specific comment, along with our modifications:
> ***
> ***Weakness 1***:
> We sincerely thank the reviewer for identifying the weaknesses in our writing. We have reorganized and rewritten the entire main text. We abandoned our previous writing approach and instead present each part of our work step-by-step, following our actual research process, and explain the motivation of each part. Please allow me to briefly introduce our research motivation, content, contributions, and the rationale behind our introduction section:
>
> >Current development of LLMs overly relies on increasing model parameters and data volume, making the development costs increasingly prohibitive, which is an unscientific and unsustainable path (Introduction, Paragraph 1). To break this dilemma, a key solution lies in understanding how current models actually learn and operate. This is also the shared goal of numerous interpretability works. Among these, many studies have focused on how the embedding space encodes natural language, finding that the embedding space can learn semantic associations between tokens—for instance, digit embeddings are often arranged following numerical order (Figure 1). Our research objective is to provide an analytical framework that theoretically clarifies why embeddings can learn these associations, with corresponding experimental verification.
> >
> >Through analyzing the gradient flow of embeddings, we discovered how token relationships affect embedding dynamics (Proposition 1). Simultaneously, we found that which token relationships embeddings capture is concretely linked to model architecture. We separately analyzed linear models (Corollary 1) and feedforward networks (Corollary 2), and verified our analysis through controlled experiments. Furthermore, we extend our analysis framework to Transformer-based LM (Section 6). Even with the leading term of the model, the probability signature we uncover could largely dominate the dynamics of the embedding structure. Our key contribution lies in proposing a viable analytical method for embedding structures. Researchers can follow our approach to conduct similar studies on models of their interest, to analyze which token relationships influence the embedding space under that specific model architecture.
> ***
> ***Weakness 2***: We thank the reviewer for this valuable feedback. We have streamlined the main text by moving all supplementary experimental analyses to the appendix, retaining only key results and insights, which significantly improves readability. Additionally, we have reformulated Proposition 1 and Proposition 2 using expectation notation, reducing mathematical symbol complexity. Every symbol is now clearly defined, and we precede the introduction of gradient flow in Section 4 with a dedicated explanation to ensure accessibility for readers.
> ***
> **Weakness 3 & Question 3**: We thank the reviewer for this valuable feedback. Our contribution lies in proposing a research methodology for analyzing how embedding spaces capture semantic relationships between tokens under different model architectures. We discuss the relationships between our work and further application in detail in Section 7 as follows:
>
> >We illustrate that each architecture implicitly selects which probability signatures it can encode. Our gradient-flow analysis makes this selection explicit and quantifiable: Corollary 1 proves that linear models cannot encode joint token-label relationships ($ \phi_x^{X\mid y} $). Any task requiring this relationship will fail, regardless of scale. Adding a nonlinear activation unlocks $\phi_x^{X\mid y}$ (Corollary 2), enabling models to learn such semantics. This suggests a principled architecture search: introduce modules whose Jacobians $G^{(1)}$ encode desired probability signatures. On the other hand, our results have shown that the loss function is not merely a performance metric but also a gradient flow sculptor that determines which probability signatures dominate. Corollary 4 shows that next-token prediction makes $\phi_s^{\rm next}$ the dominant signature, embedding tokens based on immediate neighbors. This explains why standard autoregressive models excel at local coherence but struggle with long-range dependencies. If the loss predicts $k$ future tokens, gradient flow will encode the k-gram relationship distribution. This provides a theoretical explanation for why multi-token prediction could easily capture the global relationships (Gloeckle et al., 2024).

---

> > ### Author Response · Authors · 2025-11-19
> > **Reply to Reviewer sy2v (2/2)**
> >
> > ***Weakness 4***:Our contribution does not lie in proposing specific probability signatures, but rather in introducing this research methodology of analyzing embedding structures through gradient flow equations. Through analysis of simplified models and experimental validation, we demonstrate that this research framework is both feasible and accurate. For real language models, by analyzing only the dominant term in the residual stream, the resulting probability signature can already capture a substantial portion of the embedding structure, including the ordered arrangement of digits shown in Figure 1. This precisely proves that our analytical approach is sufficiently effective and reasonable. For more complex model frameworks, we will conduct specific analyses in future work, and researchers can also follow our approach to perform similar analyses on model architectures of their interest. We added this motivation in the revised manuscript in Section 6 as follows:
> >
> > >A full analysis of all terms in Proposition 1 for Transformers would be intractable and, more importantly, unnecessary for validating our core contribution. We therefore adopt a minimalist validation strategy: analyze the dominant probability signature predicted by gradient flow and test whether it alone can predict embedding structure. If this simplified analysis succeeds, it proves that our framework captures the essential mechanism and researchers can then extend it to additional modules as needed.
> > ***
> > ***Weakness 5 & Question1,2***:Articles [1] discuss embedding structures under different initialization scales, demonstrating that small initialization scales better facilitate the capture of meaningful semantic associations in embedding spaces. Moreover, our analytical framework is not limited to linearized components: our analysis of feedforward networks precisely demonstrates how non-linear activation functions operate, with results showing that non-linear activations can capture more complex token relationships. Additionally, we discovered in the addition tasks that the model's embedding structure consistently maintains the final form shown in Figures 3 and 4 throughout training until stabilization. Related work has similarly found that semantic structures learned by language models in early stages persist into later training phases [2,3,4]. We continue the training to 20 epochs in each experiment in Section 6 and provide a dynamics evaluation in the revised version.  In Figure 6, we demonstrate the evolution of correlation between embedding structure and probability signatures during language model training, confirming that models capture semantic associations early and maintain them consistently. We have added the following analysis in Chapter 6:
> >
> > >Figure 6 B tracks the $R_{\cos}\left(W^E,\phi^{\rm next}\right)$ and $R_{\rm cos}\left(W^U,\varphi^{\rm pre}\right)$ across all subsets during training. Correlations increase during the first epoch, indicating that gradient flow rapidly encodes next-token and previous-token statistics into embeddings and unembeddings. After reaching peak alignment, correlations plateau and dip slightly, showing that the embedding structure is still largely impacted by $\phi^{\rm next}_s$ and $\varphi^{\rm pre}_s$. The fact that a single simplified probability signature maintains predictive power throughout training, proves that our gradient flow analysis captures the essential mechanism of embedding structure.
> >
> > [1] Zhongwang Zhang, Pengxiao Lin, Zhiwei Wang, Yaoyu Zhang, Zhi-Qin John Xu, Initialization is Critical to Whether Transformers Fit Composite Functions by Reasoning or Memorizing, NeurIPS 2024.
> >
> > [2] Catherine Olsson, Nelson Elhage, Neel Nanda, Nicholas Joseph, Nova DasSarma, Tom Henighan,
> > Ben Mann, Amanda Askell, Yuntao Bai, Anna Chen, et al. In-context learning and induction
> > heads. *arXiv preprint arXiv:2209.11895*, 2022.
> >
> > [3] Nelson Elhage, Neel Nanda, Catherine Olsson, Tom Henighan, Nicholas Joseph, Ben Mann,
> > Amanda Askell, Yuntao Bai, Anna Chen, Tom Conerly, et al. A mathematical framework for
> > transformer circuits. *Transformer Circuits Thread*, 1(1):12, 2021.
> >
> > [4] Neel Nanda, Lawrence Chan, Tom Lieberum, Jess Smith, and Jacob Steinhardt. Progress measures
> > for grokking via mechanistic interpretability. *arXiv preprint arXiv:2301.05217*, 2023.
> > ***
> > We sincerely thank you once again for your invaluable comments. We hope that our revisions and responses could address your concerns. We wish you the very best.

---

### Author Response · Authors · 2025-11-29
**Brief summary of the rebuttal [1/2]**

Dear AC,

Thank you for your time and effort in serving our community. In an effort to facilitate your review process, we have provided a concise summary of the key motivations, contributions, and technical content of our work below.

> Current development of LLMs overly relies on increasing model parameters and data volume, making the development costs increasingly prohibitive, which is an unscientific and unsustainable path. To break this dilemma, a key solution lies in understanding how current models actually learn and operate. This is also the shared goal of numerous interpretability works. Among these, many studies have focused on how the embedding space encodes natural language, finding that the embedding space can learn semantic associations between tokens—for instance, digit embeddings are often arranged following numerical order (Figure 1). Our research objective is to provide an analytical framework that theoretically clarifies why embeddings can learn these associations, with corresponding experimental verification.
>
>
> Through analyzing the gradient flow of embeddings, we discovered how token relationships affect embedding dynamics (Proposition 1). Simultaneously, we found that which token relationships embeddings capture is concretely linked to model architecture. We separately analyzed linear models (Corollary 1) and feedforward networks (Corollary 2), and verified our analysis through controlled experiments. Furthermore, we extend our analysis framework to Transformer-based LM (Section 6). Even with the leading term of the model, the probability signature we uncover could largely dominate the dynamics of the embedding structure. ***Our key contribution lies in proposing a viable analytical method for embedding structures. Researchers can follow our approach to conduct similar studies on models of their interest, to analyze which token relationships influence the embedding space under that specific model architecture.***

The reviewers acknowledged the value of our work, describing it as ***interesting and ``full of good ideas’’*** (sy2v, qPqX, khiJ). They also raised several constructive critiques and questions, to which we have provided detailed responses and implemented corresponding revisions in the manuscript. The key points of this discussion are summarized as follows:

- **The writing of the manuscript needed significant improvement.** (sy2v, qPqX, PGHA, khiJ)

We sincerely thank the reviewers for identifying the weaknesses in our writing. We have reorganized and rewritten the entire main text. We abandoned our previous writing approach and instead present each part of our work step-by-step, following our actual research process, and explain the motivation of each part.

We have moved all supplementary experimental analyses to the appendix, retaining only key results and essential discussions in the main text. This significantly improves readability for readers. Additionally, we rewrote Propositions 1 and 2 in expectation form, which substantially reduces mathematical notation complexity. Every symbol is now clearly defined upon first use, and we added a dedicated introduction to gradient flow concepts in Section 4 before presenting our formal analysis. These changes collectively make the paper much more accessible while preserving its technical rigor. We believe our revisions have successfully addressed the concerns regarding clarity. We are confident that the revised manuscript now presents our work with significantly improved readability and effectively communicates its contributions.

- **Lack of discussion on potential applications.** (sy2v, qPqX)

We thank the reviewers for this valuable feedback. Our contribution lies in proposing a research methodology for analyzing how embedding spaces capture semantic relationships between tokens under different model architectures. We added a discussion of the relationships between our work and further application in detail in Section 7 (lines 514-527).  We discussed how our approaches guide the design of model architecture and loss functions.

- **How do some of the metrics progress across training?** (sy2v, qPqX)

We agree that the dynamics of the evaluation metrics are essential. We continue the training to 20 epochs in each experiment in Section 6 and provide a dynamics evaluation in the revised version. In Figure 6, we show the evolution of the correlation between the embedding structure and the probability signature structure throughout the whole training, and provide an analysis in the revised manuscript (lines 450-458).  ***The fact that a single simplified probability signature maintains predictive power throughout training, proves that our gradient flow analysis captures the essential mechanism of embedding structure.***

---

> ### Author Response · Authors · 2025-11-29
> **Brief summary of the rebuttal [2/2]**
>
> - **Extending the Analysis to Complex Models.** (sy2v, qPqX)
>
> Our contribution does not lie in proposing specific probability signatures, but rather in introducing this research methodology of analyzing embedding structures through gradient flow equations. Through analysis of simplified models and experimental validation, we demonstrate that this research framework is both feasible and accurate. For real language models, by analyzing only the dominant term in the residual stream, the resulting probability signature can already capture a substantial portion of the embedding structure, including the ordered arrangement of digits shown in Figure 1. This precisely proves that our analytical approach is sufficiently effective and reasonable. For more complex model frameworks, we will conduct specific analyses in future work, and researchers can also follow our approach to perform similar analyses on model architectures of their interest. We added this motivation in the revised manuscript (lines 416-421).
>
> - **The Contribution of our work is not clear.** (PGHA, khiJ)
>
> We acknowledge that the initial presentation may have led to some misunderstandings regarding our core contributions. We have thoroughly restructured the manuscript and added explicit statements of our contributions in multiple sections to ensure they are now clearly and unambiguously presented. We are confident that these revisions successfully clarify any previous ambiguities.
>
> - **Relationship with the semantics.**(PGHA)
>
> In the revised manuscript, we provide a detailed discussion in Section 3.2 on the connection between the token relationships we investigate, our defined probability signature, and semantics (lines 152-167). For each probability signature, we present concrete examples to demonstrate the specific semantic concepts it captures.
>
> - **Why small initialization.**(sy2v)
>
> Article [1] discussed embedding structures under different initialization scales, demonstrating that small initialization scales better facilitate the capture of meaningful semantic associations in embedding spaces. In the NTK regime (large initialization scale), the embedding structure may fail to capture token relationships. In the revised version, we clarify this in lines 488-490.
>
> [1] Zhongwang Zhang, Pengxiao Lin, Zhiwei Wang, Yaoyu Zhang, Zhi-Qin John Xu, Initialization is Critical to Whether Transformers Fit Composite Functions by Reasoning or Memorizing, NeurIPS 2024.
>
> ---
>
> Thank you for your efforts. We kindly ask you to consider the revised manuscript following our rebuttal, and we look forward to your feedback.
>
> Best regards,
>
> Authors.

---

### Meta-Review · Area_Chair_4DH2 · 2026-01-13

**Summary:**

This submission proposes a link between embedding geometry and token relationships in language models via gradient flow dynamics. The presented results are interesting, and have potential, but the limited evaluation (both theoretical and empirical) has much room for improvement. Accessibility and writing was also a common concern.

**Reviewer Concerns:**

- sy2v writting: **No**
- sy2v clearness and rigorousness: **No**
- sy2v practical implications: **No**
- sy2v simplified theoretical analysis: **No**
- sy2v small initialization: **No**
- qPqX clarity and organization: **No**
- qPqX large-scale experiments: **No**
- qPqX multiple transformer blocks: **No**
- qPqX practical implications: Yes
- PGHA challenging to read: **No**
- PGHA novelty: **No**
- PGHA semantics: **No**
- khiJ clarity: **No**
- khiJ mathematical accuracy: **No**

**Reviewer Scores:**

sy2v 4->4
qPqX 4->4
PGHA 2->2
khiJ 4->4

---

### Decision · Program_Chairs · 2026-01-26

Reject